# Mapping chromatin accessibility and active regulatory elements reveals pathological mechanisms in human gliomas

Karolina Stępniak[1,9], Magdalena A. Machnicka [2,9], Jakub Mieczkowski [1,8,9], Anna Macioszek[2], Bartosz Wojtaś [1], Bartłomiej Gielniewski[1], Katarzyna Poleszak [1], Malgorzata Perycz[1], Sylwia K. Król[1], Rafał Guzik[1], Michał J. Dąbrowski [3], Michał Dramiński[3], Marta Jardanowska[3], Ilona Grabowicz[3], Agata Dziedzic[3], Hanna Kranas [2,7], Karolina Sienkiewicz[2], Klev Diamanti [4], Katarzyna Kotulska[5], Wiesława Grajkowska[5], Marcin Roszkowski[5], Tomasz Czernicki[6], Andrzej Marchel[6], Jan Komorowski [4], Bozena Kaminska [1✉] & Bartek Wilczyński [2✉]

Chromatin structure and accessibility, and combinatorial binding of transcription factors to regulatory elements in genomic DNA control transcription. Genetic variations in genes encoding histones, epigenetics-related enzymes or modifiers affect chromatin structure/ dynamics and result in alterations in gene expression contributing to cancer development or progression. Gliomas are brain tumors frequently associated with epigenetics-related gene deregulation. We perform whole-genome mapping of chromatin accessibility, histone modifications, DNA methylation patterns and transcriptome analysis simultaneously in multiple tumor samples to unravel epigenetic dysfunctions driving gliomagenesis. Based on the results of the integrative analysis of the acquired profiles, we create an atlas of active enhancers and promoters in benign and malignant gliomas. We explore these elements and intersect with Hi-C data to uncover molecular mechanisms instructing gene expression in gliomas.

[1] Nencki Institute of Experimental Biology of the Polish Academy of Sciences, Warsaw, Poland. [2] Faculty of Mathematics, Informatics and Mechanics, University of Warsaw, Warsaw, Poland. [3] Institute of Computer Science of the Polish Academy of Sciences, Warsaw, Poland. [4] Department of Cell and Molecular Biology, Uppsala University, Uppsala, Sweden. [5] Departments of Neurology, Neurosurgery, Neuropathology, The Children's Memorial Health Institute, Warsaw, Poland. [6] Department of Neurosurgery, Medical University of Warsaw, Warsaw, Poland. [7] Present address: Institute for Research in Biomedicine (IRB Barcelona), The Barcelona Institute of Science and Technology, Barcelona, Spain. [8] Present address: Medical University of Gdansk, International Research Agenda 3P Medicine Laboratory, Gdansk, Poland. [9] These authors contributed equally: Karolina Stępniak, Magdalena A. Machnicka, Jakub Mieczkowski. ✉email: b.kaminska@nencki.edu.pl; b.wilczynski@mimuw.edu.pl

Gliomas are most common primary tumors of the central nervous system (CNS). According to a World Health Organization (WHO) classification, gliomas are divided based on molecular dysfunctions and histopathology into WHO grade I gliomas (benign tumors with infrequent genomic alterations and long-term survival), grades II and III diffuse gliomas (characterized by the presence or absence of mutations in *IDH*, *TERT*, *ATRX* and copy number alterations of 1p, 19q, 7, and 10q) and grade IV glioblastoma, the most malignant tumor with numerous genomic alterations[1,2]. Gliomagenesis and progression are associated with accumulation of genetic and epigenetic alterations[3]. Comprehensive sequencing studies have revealed numerous aberrations in epigenetics- and chromatin remodeling-related genes in high grade gliomas. Due to aberrant activation or inactivation of epigenetics-related enzymes the epigenetic landscape is highly deregulated leading to aberrant gene expression[4–6].

Chromatin acts as a dynamic platform of signal integration and a long-term maintenance of gene expression profiles. Modifications of histones, particularly their N-terminal tails, influence chromatin accessibility and gene expression, and can be activating or repressive[7]. Acetylation of N-terminal lysine residues of histones H3 and H4, trimethylation of H3 lysine 4 (H3K4me3) and lysine 79 (H3K79me3), and trimethylation of H3 lysine 36 (H3K36me3), are typically associated with an active chromatin, while methylation of lysines 9 (H3K9me3) and 27 (H3K27me3) of histone H3 are hallmarks of a condensed chromatin at silent loci[8]. H3 acetylation and H3K4me1 marks outside of promoter regions have been correlated with functional enhancers in different cell types[9,10]. The deceptively opposite modifications, H3K4me3 and H3K27me3 co-localize in regions termed "bivalent domains", mostly in embryonic stem cells[11]. Extensive characterization of histone methylations has been completed at a limited genome coverage and resolution in selected human cells[12]. There are only a few reports regarding profiles of histone modifications or chromatin accessibility in freshly resected human specimens. This includes high-resolution Hi-C maps of chromatin contacts in human embryonic brain[13], ATAC-seq on postmortem brain[14] or glioma stem-like cells from glioblastoma[15], ChIP-seq analysis for H3K27me3 modification on 11 glioblastoma and 2 diffuse astrocytoma samples[16]. Only recently researchers began integrating H3K27ac profiles with gene expression, DNA methylomes, copy number variations, and whole exomes demonstrating distinct chromatin and epigenetic profiles in glioblastomas, distinguishing this entity from other brain tumors[17]. Combination of those features in a single patient and intersections of acquired profiles in gliomas of different grades have not yet been achieved.

We performed a comprehensive analysis of whole genome profiles of open chromatin, histone modifications, DNA methylation and gene expression in tissues of patients using freshly resected gliomas of various malignancy grades. Intersecting the acquired data led to more precise mapping of active promoters and enhancers in gliomas, some reflecting brain-specific sites. We intersected our data with Hi-C maps from a human embryonic brain to identify topologically associated domains and we searched for co-regulated genes, regulatory binding motifs and potential master regulators. This approach revealed a regulatory connection between FOXM1 and ANXA2R implicated in gliomagenesis. We provide an atlas of brain specific enhancers and promoters which could be a valuable resource for further exploration. The data are available at http://regulomics.mimuw.edu.pl/GliomaAtlas/ in raw and processed forms ready to visualize in genome browsers.

## Results

**Creating the atlas of regulatory sites in benign and malignant gliomas.** We focused on groups of patients representing major malignancy groups: pilocytic astrocytomas (WHO grade I, PA, $n = 11$), diffuse astrocytomas (WHO grade II and III, DA, $n = 7$) and glioblastomas (WHO grade IV, GBM, $n = 15$). Clinical data about each patient have been collected and are presented in Supplementary Table 1. Freshly resected glioma tissues were immediately homogenized to a single cell suspension and experimentally defined numbers of cells were split into portions that were directly processed for ATAC-seq, ChIP-seq, RNA-seq, or DNA methylation experiments (Fig. 1). Tissue homogenization was performed to avoid confounding factors related to tumor heterogeneity. We analyzed the profiles of DNA methylation, open chromatin and histone modifications characteristic for active (H3K4me3, H3K27ac) and repressed chromatin (H3K27me3) along with RNA-sequencing on the individual patient tissues (the number of patient samples for each protocol is available in Supplementary Table 2). The *IDH1/2* mutational status has been evaluated in all samples by Sanger sequencing of PCR amplicons (Supplementary Table 1).

For each individual, we performed assay for transposase-accessible chromatin with sequencing (ATAC-seq) and chromatin immunoprecipitation followed by sequencing (ChIP-seq) to detect histone modification marks characteristic for active and poised enhancers, and active promoters. Figure 2A summarizes numbers of samples for which ATAC-seq and ChIP-seq were collected excluding the samples that did not meet our stringent quality criteria. For a number of glioma samples we collected complete datasets, except for a lack of H3K27me3 profiles from PAs, due to a low size of the resected tumors (Supplementary Table 2). Representative ATAC-seq, ChIP-seq and RNA-seq peaks for the *NRXN2* gene in two samples, one selected from pilocytic astrocytomas (PA04) and one from glioblastomas (GB01) are shown in Fig. 2B. The peak enrichments are comparable across samples with expected overrepresentation of peaks in promoter regions. Numbers of detected peaks are presented in Fig. 2C.

By performing ChIP-seq experiments for two active and one repressive histone marks (H3K4me3, H3K27ac, and H3K27me3), ATAC-seq and RNA-seq we have collected a comprehensive dataset describing chromatin state and gene activity across different glioma grades. We identified regulatory regions whose activity is shared between samples ("common") or specific to a small subset of samples ("variable") for both H3K4me3 and H3K27ac ChIP-seq experiments (Fig. 2D). We identify many more common than variable H3K4me3 peaks, suggesting those commonly active promoters represent genes expressed in the brain, while the variable ones may be more related to gliomagenesis.

Genome-wide chromatin conformation capture (Hi-C) allows identification of interactions between DNA regions including promoter–enhancer interactions. To better describe enhancers, we used Hi-C data from high-resolution 3D maps of chromatin contacts from developing human brain[13] for identification of enhancer–promoter contacts that might function in gliomas. First, using Hi-C data we searched for contacts of all putative active enhancers identified in our samples with any region within 2M bp range (Fig. 2E). We found significant ($p < 0.001$) contacts for 21,141 putative enhancers out of 45,548 (including 5,530 common enhancers out of 10,673). Target genomic intervals that were predicted to be in contact with common putative enhancers are enriched in promoter regions, supporting biological relevance of these predictions (Supplementary Fig. 2).

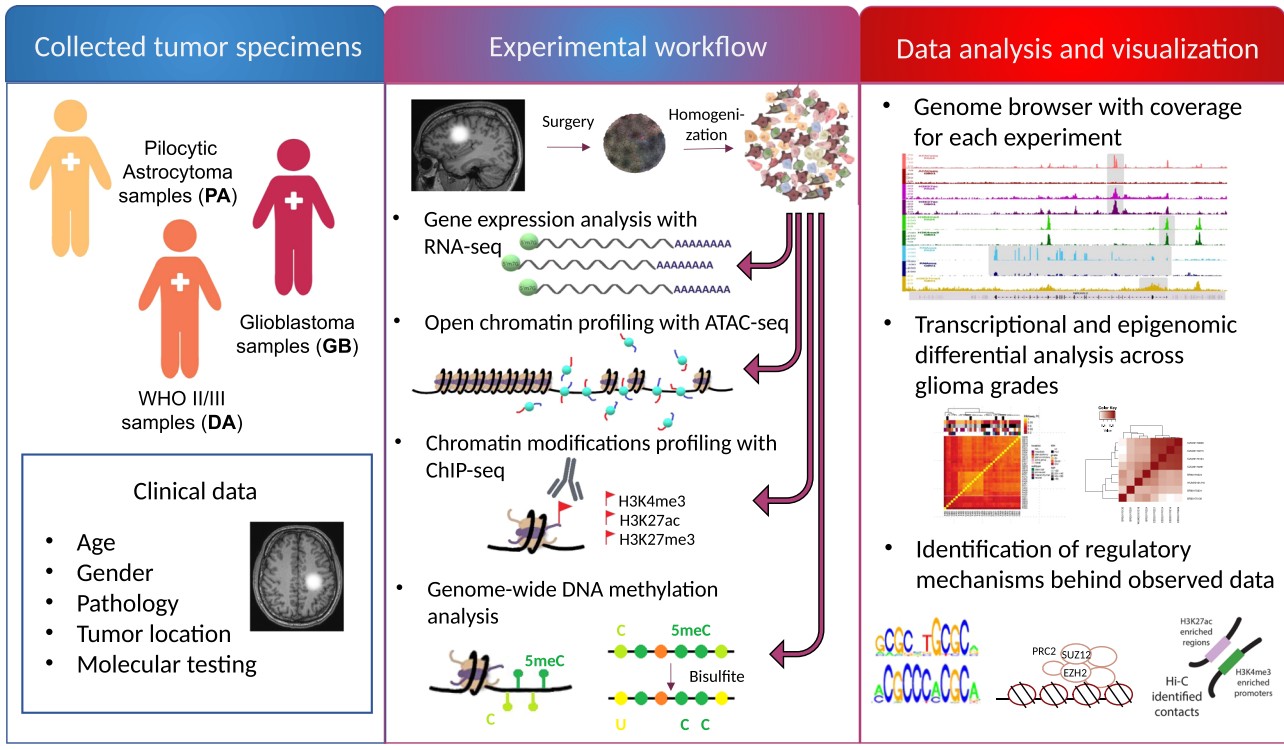

**Fig. 1 Experimental workflow.** Freshly resected glioma specimens were acquired from patients representing major glioma malignancy groups: pilocytic astrocytomas (WHO grade I, PA, $n = 11$), diffuse astrocytomas (WHO grade II and III, DA, $n = 7$) and glioblastomas (WHO grade IV, GBM, $n = 15$). Clinical data about each patient was collected. Glioma tissues were homogenized to a single cell suspension and experimentally defined numbers of cells were split into portions that were processed for RNA-seq, ATAC-seq, ChIP-seq, or DNA methylation experiments. Differential analysis of epigenomic and transciptomic profiles across glioma grades was performed, followed by identification of related regulatory mechanisms. Results from all genome-wide experiments were made available in a genome browser (http://regulomics.mimuw.edu.pl/GliomaAtlas/).

**Identified, non-promoter regulatory regions show high diversity between patients**. Active promoters, defined by the presence of either H3K4me3 or H3K27ac, are highly consistent between patients. Similarly, gene expression profiles show significant overlap between patients (Fig. 3A, Supplementary Fig. 3). Expectedly, sets of promoters identified as active based on the presence of either one of the active histone marks or open chromatin peaks, are highly similar; 8,007 common active promoters display ATAC-seq signals, H3K4me3 and H3K27ac marks (Fig. 3B). Active promoters exhibit also evolutionary conservation in the context of 100 vertebrate genomes, compared to random genomic intervals (Fig. 3C). In contrast to promoter regions, putative enhancers show high variability between patients (Fig. 3A, Supplementary Fig. 3). Their sequence conservation is also weaker than sequence conservation of the promoters, but still very significant when compared to random genomic intervals (Fig. 3C). By comparing these common and variable promoter regions to the published studies of promoters active in normal brain (Gerstein et al. [18]) and in non-brain cell lines, we can estimate that ~50% of common promoters are also active in normal brain and 30% are also active in other tissues (Supplementary Fig. 3B). The values for variable elements are considerably lower, with ~90% of them not appearing in brain or cell-type derived datasets, making them likely to be specific to gliomas.

Having paired ChIP-seq and RNA-seq data from the same samples we investigated whether the expected correlation between the level of active histone marks and gene expression levels[19] is visible in our data. Both H3K4me3 and H3K27ac ChIP-seq coverages at promoter regions correlate with the expression of the associated transcripts (Spearman R = 0.62 and 0.66,

respectively; Fig. 3D). However, our data demonstrate that dependency between occurrence of H3K4me3 peaks and transcript expression is quite non-linear, with a pronounced difference in coverage between transcripts whose expression is within 1st or 2nd quantile. In contrast, in case of H3K27ac the increase in the transcript expression is associated with the steady increase in the ChIP-seq signal (Fig. 3D).

Hierarchical clustering on the mRNA expression profiles of the samples revealed that malignancy groups correspond very well to the obtained clusters (Fig. 3E). All but one PA samples formed a distinct cluster showing a stronger within-group similarity than groups related to other malignancies. As expected due to the presence of a PA-like glioma subtype within the LGr4 RNA cluster[1], majority of the PA samples were classified to LGr4 Pan Glioma RNA Expression Cluster. DA samples also exhibited a high similarity of RNA expression profiles. Majority of them were assigned to LGr1/LGr3 RNA clusters. In concordance with Pan Glioma RNA Expression Cluster specificity, all *IDH1* mutant samples were assigned to the LGr1 or LGr3 RNA clusters, even though they were split on the dendrogram into DA and GB specific clusters. The GB mRNA expression profiles exhibited the largest variability, although the majority of them were found to be of a Mesenchymal glioblastoma subtype assigned to the LGr4 RNA cluster.

To further assess biological relevance of enhancer–promoter contacts predicted in glioma samples, we used Hi-C data from a fetal human brain[13]. We computed the Spearman correlation of H3K27ac coverage on enhancers with transcript expression for enhancer–promoter pairs defined either based on Hi-C data or by assigning the closest transcript to each analyzed enhancer. We found that both for closest enhancer–transcript pairs and pairs

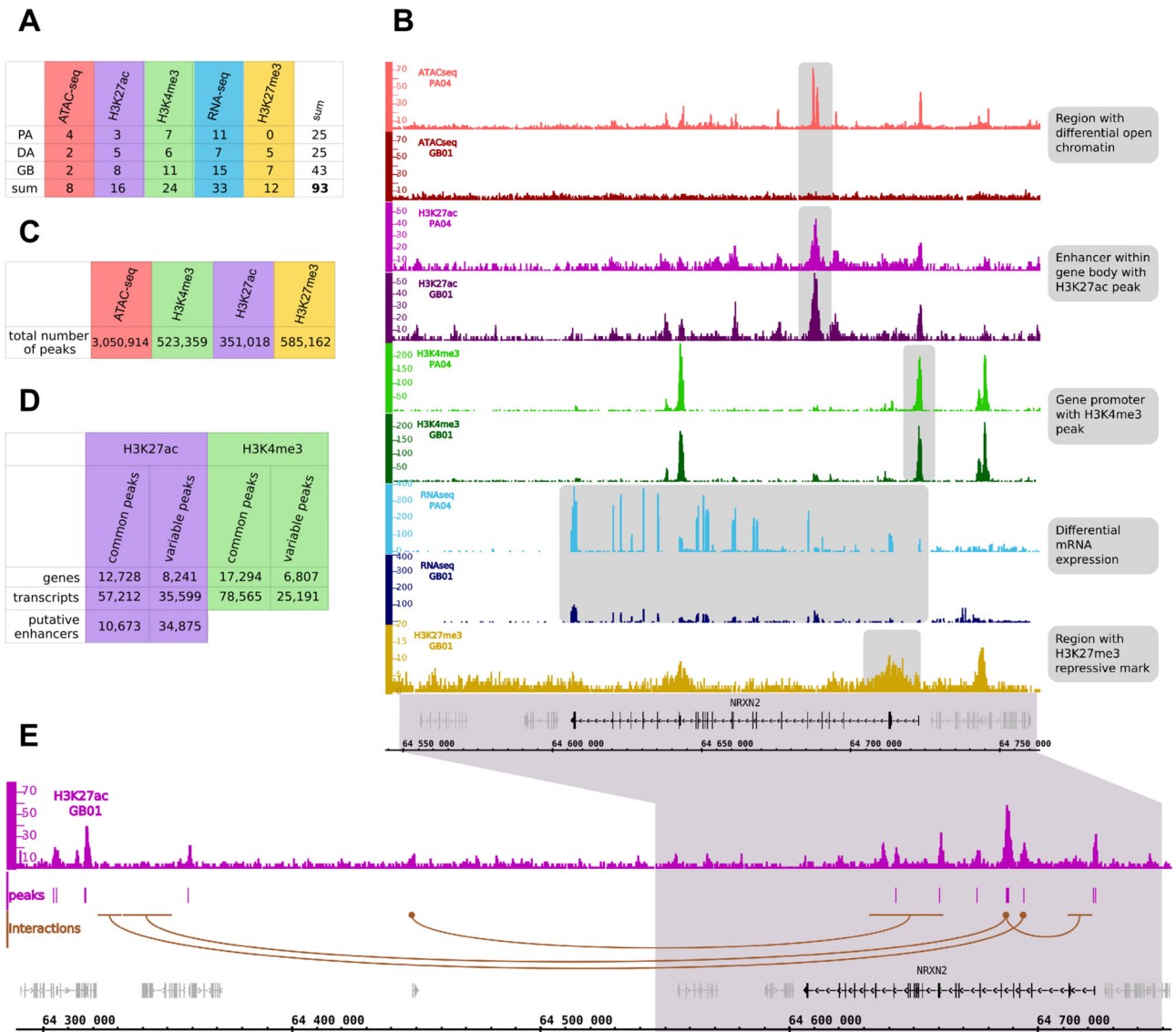

**Fig. 2 Construction of an atlas of regulatory sites in gliomas. A** Numbers of samples from gliomas of different grades for which ATAC-seq, ChIP-seq, and RNA-seq data were collected. **B** Representative peaks of ATAC-seq, ChIP-seq, RNA-seq at the *NRXN2* gene in pilocytic astocytomas (PA04) and glioblastoma (GB01) samples. Note high expression of the *NRXN2* in PA in comparison to GB, which correlates with the lack of the ATAC-seq signal in GB (shadowed blocks). **C** Total numbers of peaks identified in all samples across performed experiments. **D** Numbers of identified common and variable active regulatory elements, genes and transcripts. **E** Example of promoter-enhancer interactions identified using Hi-C data (Won et al., 2016). Brown dots represent "anchors", while brown horizontal lines depict predicted interaction sites.

identified from Hi-C data the correlation between the histone mark level and transcript expression is relatively low (around 0.1–0.25) (Fig. 3F). Our results suggest that closest enhancer–promoter pairs are expected to be correlated if the distance between them is at most 20 kb. However, in cases when no transcript can be found within 200K bp from the putative enhancer, Hi-C data can provide significantly more information about expected enhancer–promoter interactions than assignment based on the shortest promoter–enhancer distance.

Altogether, the presented results on the integrative analysis of ATAC-seq, ChIP-seq, and RNA-seq data allowed us to identify non-coding, regulatory sites that are active in gliomas and represent common brain specific sites. We predicted the presence of active promoters and putative enhancers, and predicted enhancer–promoter contacts in glioma samples by exploiting the published Hi-C data dataset[13]. The data allowed us to create an atlas of non-coding, regulatory sites in those brain tumors and

we made the data easily available in raw and processed formats in a genome browser on our website (http://regulomics.mimuw.edu.pl/GliomaAtlas).

**H3K27ac at the transcription start sites marks genes related to neuronal signaling in pilocytic astrocytomas.** We explored the atlas of active promoters and putative enhancers to identify tumor–specific gene regulation. We compared chromatin activity around transcription start sites (TSS) between benign (PA) and malignant (DA/GB) tumors using H3K4me3 and H3K27ac profiles (Fig. 4A, B). The analysis was limited to protein coding genes. We found that the abundance of the selected histone marks discriminated the two glioma groups. While H3K4me3 marks at TSS were significantly higher in PA samples (Fig. 4A), H3K27ac signals were significantly more abundant in DA/GB samples (Fig. 4B, Supplementary Fig. 4A).

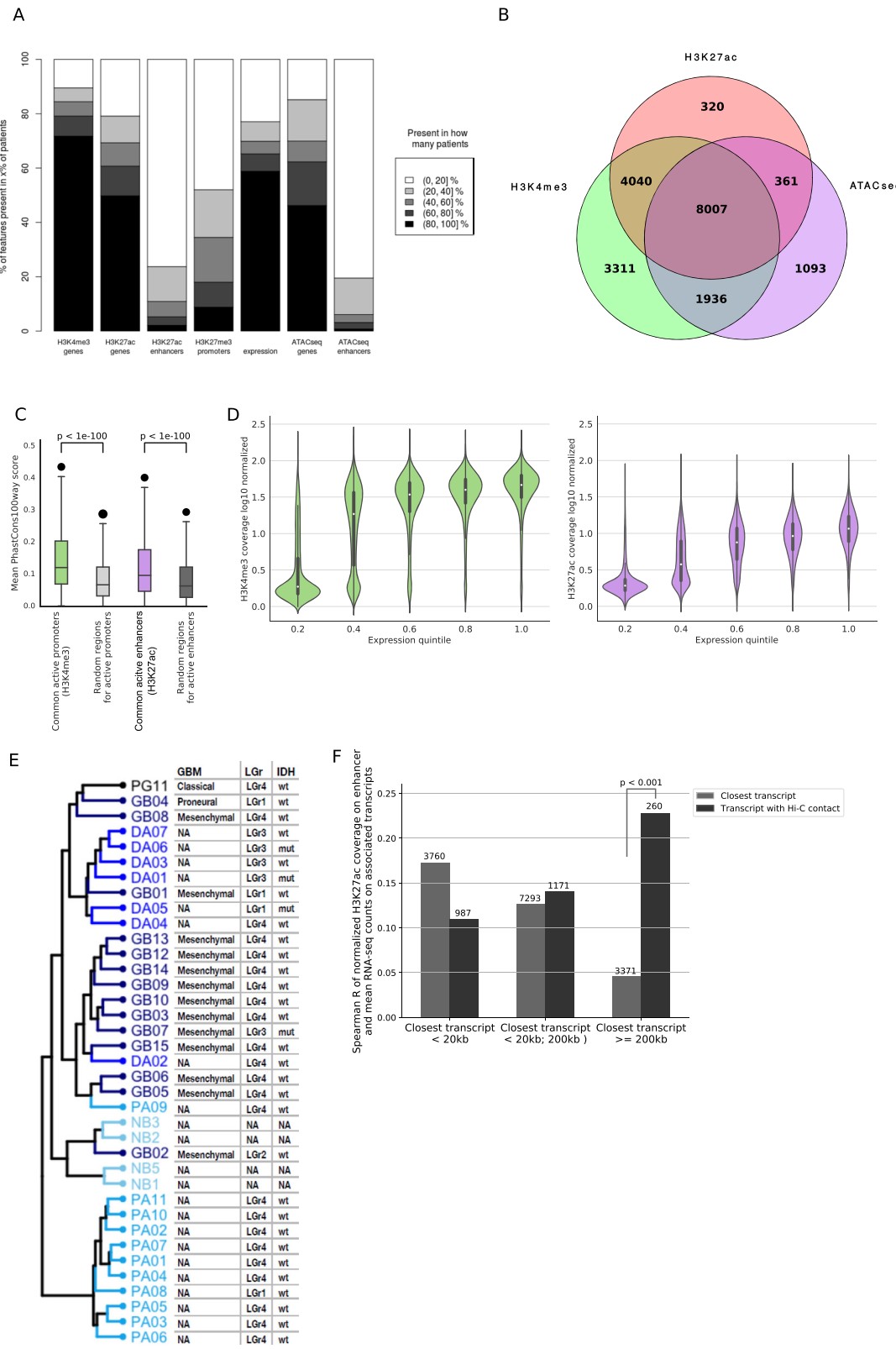

Interestingly, besides the global increase of H3K27ac marks in DA/GB, we found a set of regions with higher H3K27ac abundance in PA than in DA/GB samples (Fig. 4C, we will refer to them as the selected regions in the following paragraph). The detailed H3K27ac profiles centered at the corresponding TSS confirmed that the observed increase in their abundance was not an artifact resulting from data transformation, and was spread over the selected region (TSS+/−2kb) (Fig. 4D). Also, ATAC-seq profiles obtained for four PA and four DA/GB samples showed differences in chromatin accessibility (Fig. 4E). To explore those changes in the genome-wide context, we compared the H3K27ac profiles in the selected regions with the remaining regions. We found that in comparison to the other regions, H3K27ac profiles in the selected regions were substantially lower in DA/GB

**Fig. 3 Global characterization of chromatin structure and its relationship with gene expression. A** Commonality of histone marks, open chromatin regions and gene expression patterns between samples. Color scale represents the percentage of patients sharing a particular genomic element status. Bar height represents a number of genomic elements (scaled to 100%). **B** Venn diagram shows intersection of the promoter regions marked by H3K4me3, H3K27ac and open chromatin sites detected with ATAC-seq. **C** Evolutionary conservation of the identified promoters (green) and enhancers (purple) measured by PhastCons 100-Way scores. Mean scores for the identified regulatory regions are compared to mean scores obtained for random genomic intervals. *P* values were calculated with one-sided Mann–Whitney *U* test. Data are represented as boxplots in which the box shows the quartiles of the dataset, the middle line is the median and the whiskers extends to the largest or smallest value no further than 1.5 × the inter-quartile range. **D** Correlation between H3K4me3 (green), H3K27ac (purple) and transcript expression. Normalized ChIP-seq coverage is plotted against an expression level of transcripts of protein coding genes, divided into quantiles. Data are represented as violin plots with a nested boxplot, where shape indicates the distribution of data. The box shows the quartiles of the dataset, the middle white dot is the median, whiskers extend to 1.5 × IQR past the low and high quartiles. **E** Hierarchical clustering of samples based on gene expression profiles. Shades of blue indicate tumor grade (from light = normal brain to dark = Grade IV and black PG). In the table on the right side the subtype of glioblastoma samples (GBM) is reported as well as assignment of glioma samples to one of Pan Glioma RNA Expression Clusters (LGr) as well as *IDH1* gene mutation status (IDH). **F** Correlation between H3K27ac ChIP-seq coverage on enhancers and transcript expression. Numbers above bars indicate numbers of enhancers in each group. The *p* value has been estimated empirically as explained in "Methods".

samples (Fig. 4F, right panel), while in PA samples, the H3K27ac levels remained unchanged (Fig. 4F, left panel). This shows that H3K27ac mark in the selected regions was diminished in samples of the malignant tumors, which may result from the absence of some regulatory modes or cell types in DA/GB tumors.

To further characterize PA-specific H3K27ac profiles in terms of gene functionality, we analyzed expression levels of the genes corresponding to the selected TSS proximal regions. The average expression levels of these genes were significantly higher (*P* = 2.8e-167, Wilcoxon rank-sum test) in PA tumors, which supported previous observations (Fig. 4G). Gene Ontology (GO) functional enrichment analysis of the selected genes indicated their participation in synaptic signaling and nervous system development (Fig. 4H, Supplementary Table 3). A search for transcription factor binding site motifs within the selected regions revealed the enrichment of factors involved in neuronal processes, including NRF1 (Fig. 4I, see "Methods" for details). NRF1 is involved in neurite growth and its deletion leads to neuronal dysfunctions[20,21]. Another identified factor, MXI1, was reported as a tumor suppressor negatively regulating c-MYC and an inhibitor of glioma proliferation[22,23]. Altogether, these findings indicate higher expression of genes related to neuronal signaling in PA tumors when compared to DA/GB gliomas.

**Pilocytic astrocytomas show H3K4me3 hypermethylation at the promoters targeted by PRC2.** Hierarchical clustering of samples with H3K4me3 marks in promoters shows a noticeable difference between PA and DA/GB gliomas (Fig. 5A). The majority of the PA samples exhibited large pairwise overlap of H3K4me3 marked promoters (0.93–0.95), while the pairwise overlap between PA samples and DA/GB gliomas was lower. In addition, the group of DA/GB tumors showed relatively lower similarity among them. We identified 359 genes with the promoters differentially marked with H3K4me3 between the five PA samples forming the tight cluster, and DA/GB gliomas. The vast majority of them (338 out of 359) had the active chromatin mark in their promoters only in PA samples (Supplementary Table 4) and were characterized by a significant increase in the mRNA levels (Fig. 5B), H3K27ac and ATAC-seq signal intensity in PA samples in comparison to DA/GB samples (Supplementary Fig. 5A, B).

Interestingly, this promoter set is rich in binding sites of the PRC2 complex components: nearly half of them contain the binding sites for EZH2 and nearly one sixth contains the binding sites for SUZ12, based on ChIP-seq data from the ENCODE project[18]. When compared to promoters marked by H3K4me3 in all our samples, the promoters specifically active in PA are enriched in EZH2 and SUZ12 binding sites 7.2 and 2.5 times,

respectively (Fig. 5C). In addition, transcription factor motifs enrichment analysis showed that the promoters specifically active in PA are characterized by the presence of GC rich sequences, while no strongly enriched motifs were identified (Supplementary Fig. 5C). This observation is in agreement with an expected lack of strong sequence specificity in the PRC2 complex binding. Moreover, when we compared the motifs that are enriched with respect to other active promoters, we found much fewer motifs significantly enriched in the PA specific promoters than when compared to other inactive promoters. Also, comparison to motifs enriched in DA/GB-specific promoters, supports lack of PA-specificity of the GC-rich motifs (Supplementary Fig. 5D).

We acquired the whole-genome bisulfite sequencing data from the same tumor samples as discussed above. Interestingly, we found DNA methylation levels at the promoters with EZH2 binding sites to be lower in PA compared to DA/GB gliomas (Fig. 5D). This observation was corroborated by the data from the independent PA and GB dataset analyzed with HumanMethylation450 BeadChip array (Fig. 5E). Overall, these promoters may represent an epigenetic signature which differentiates PA from DA/GB gliomas and indicate that mechanisms related to chromatin repression through the PRC2 complex together with DNA methylation may play leading roles in regulating a set of genes.

Using PANTHER Overrepresentation Test[24], we searched for enriched Gene Ontology (GO) terms in a set of the PA-specific genes with EZH2 binding sites in promoters. This analysis did not reveal strong enrichments, except for terms related with developmental processes, cell differentiation, transmembrane transport processes (Fig. 5F, Supplementary Fig. 5G). For genes specifically active in PA, but without EZH2 binding site in promoters, no significantly enriched GO terms were identified. We speculate that altered expression of these genes may arise as a secondary effect, since representatives of several diverse transcription factor families (including HES, GATA, NR2E, VSX, and SIM1) can be identified among those genes that have EZH2 binding sites in their promoters.

Pilocytic astrocytomas are usually located in the cerebellum, while higher grade gliomas more often develop in cortical areas[25]. We set out to verify, whether the observed set of promoters, specifically activated in PA tumors may be connected to the common developmental origin of these tumors. We collected data on DNA methylation levels of the promoters specifically active in PA and having an EZH2 binding site in previously published samples from different brain regions[26,27]. In agreement with our prediction, these promoters have increased DNA methylation levels in the cortex compared to the cerebellum (Supplementary Fig. 5F), however the increase is not as significant as in our PA

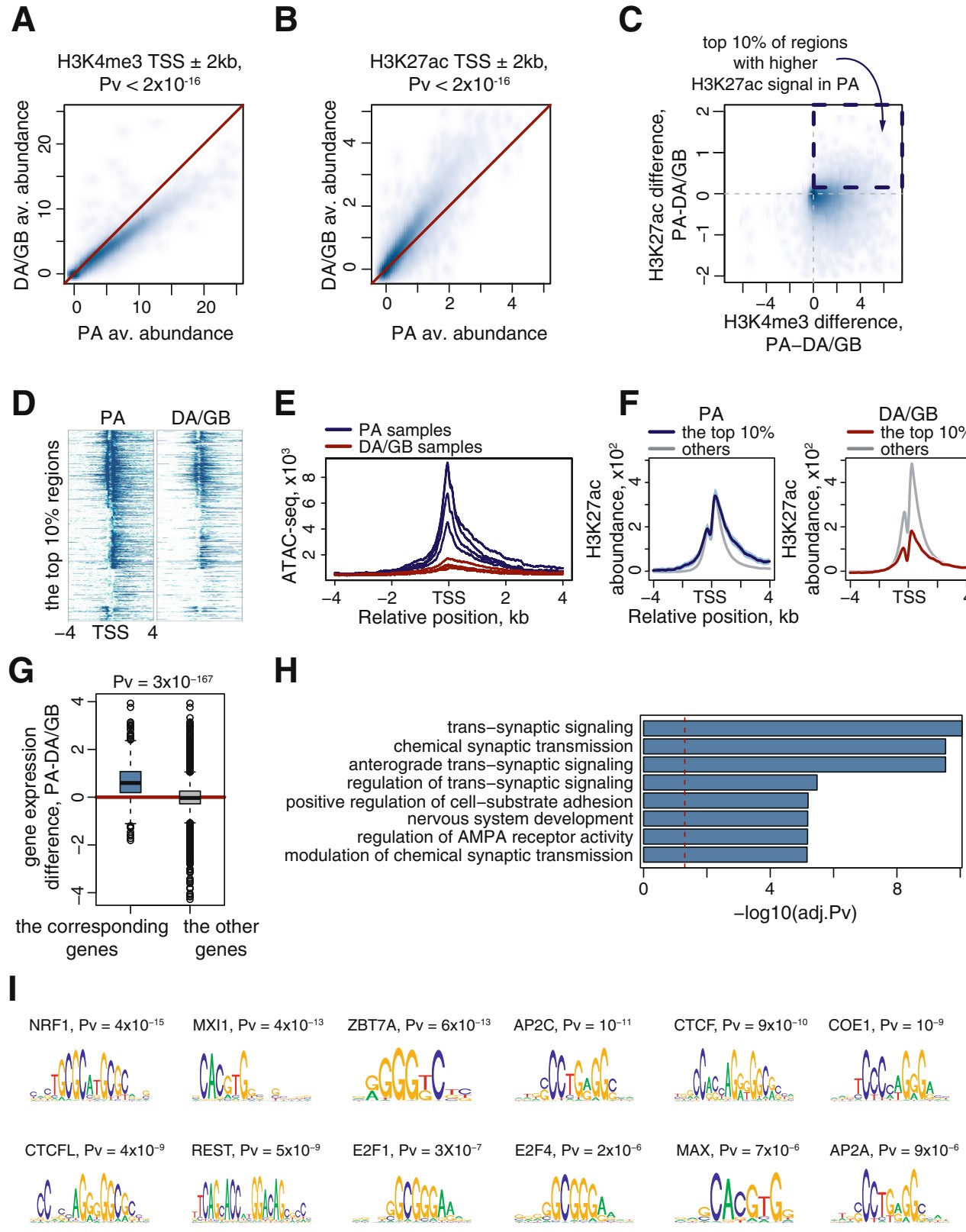

samples. Similarly, PA located in infratentorial brain regions are hypermethylated compared to the control cerebellum (Supplementary Fig. 5G). These differences in DNA methylation are accompanied by changed transcript expression levels (Supplementary Fig. 5H) supporting the functional nature of the hypermethylation. These results suggest that the epigenetic

mechanism of activation of the genes in question is a combination of their developmental background and further tumor-related dysregulation of the PRC2 complex. At the same time, these genes exhibit even higher methylation levels in GB samples, consistent with expected dysregulation of Polycomb complexes in high grade tumors.

**Fig. 4 Chromatin activity profiles indicate the presence of the normal brain signature in pilocytic astrocytomas. A**, **B** Scatter-plots representing abundance of the H3K4me3 (**A**) and H3K27ac (**B**) marks obtained for PAs (*X*-axes) and DA/GBs (*Y*-axes). A single point on the scatter-plots represents an average abundance (ChIP-input) of a corresponding histone mark around a transcription start site [TSS + /-2 kb]. *P* values shown above the plots were estimated with the Wilcoxon rank-sum test. **C** Comparison of abundance values shown on 4A (*X*-axis) and 4B (*Y*-axis). The dashed frame indicates the strongest difference in H3K27ac abundance (top 10%) values. **D** Heatmaps showing H3K27ac abundance around the TSS for the top 10% regions (each row corresponds to a TSS, sorted by signal abundance). **E** ATAC-seq profiles of 4 PA samples and 4 DA/GB samples (see "Methods" for details) around TSS overlap with the regions selected on 4C. The blue and red lines correspond to PA and DA/GB samples, respectively. **F** Average H3K27ac profiles around TSS. The blue and red lines correspond to the top 10% regions selected in 4C and were computed for the PA and DA/GB samples, respectively. The gray lines correspond to regions not selected on 4C. The thick lines correspond to the average profiles, while colored areas give reference of the confidence interval for mean (CI). **G** Differences of the gene expression levels computed either for genes corresponding to the top 10% selected regions (left boxplot) or to the rest of the regions (right boxplot). *P* value shown above the plot and indicating differential regulation was estimated with the two-sided Wilcoxon rank-sum test. Data are represented as boxplots in which the box shows the quartiles of the dataset, the middle line is the median and the whiskers extends to the largest or smallest value no further than 1.5 × the inter-quartile range. **H** Results of Gene Ontology over-representation analysis performed for the genes corresponding to the top 10% regions. The barplot shows scaled Bonferroni corrected *p* values from the one-sided Fisher's exact test and the vertical, dashed red line stands for the significance threshold (*P* = 0.05). **I** Logos of top DNA-binding motifs enriched in the top 10% regions. The enrichment was computed with regard to the rest of the analyzed regions. *P* values indicated above the logos stand for the significance of the enrichment and were computed with a tool from MEME-Suite (see "Methods" for details).

**Differential activation of regulatory regions coupled with chromatin contacts uncovers the FOXM1-ANXA2R axis operating in glioblastoma and impacting patient survival.** Chromatin conformation may influence regulation of gene expression[28,29] and topologically associating domains (TADs) segregate the genome into megabase-scale regions. Enhancers and their cognate promoters are typically located in the same TAD, even if the genomic distance between them is as large as a megabase[30]. Since we were most interested in promoter–enhancer contacts that require high-resolution maps, we used published high resolution fetal brain Hi-C data[13] to putative enhancer locations to search for potential interactions that may elicit pathogenic mechanisms. In order to validate that these data are applicable to putative glioma enhancers, we verified that the TAD boundaries in the fetal brain data are not significantly different from those derived from low-resolution glioma Hi-C data[31].

We identified 116 enhancer–promoter pairs with a significant contact frequency based on Hi-C data and a high correlation between the enhancer coverage in H3K27ac ChIP-seq data and transcript expression (Spearman R > 0.7; Fig. 6A). The identified contacts corresponded to 96 genes among which 17 (FDR < 0.01) were differentially expressed in tumors of different malignancy (Fig. 6B). We focused on genes whose expression levels were the highest in GB samples and the lowest in PA samples. We obtained a list of five genes, and for all of them we found a significant correlation between promoter and putative enhancer activity levels (Fig. 6C). One of the identified genes, *ANXA2R*, encodes a receptor for Annexin A2 (ANXA2), an element of ANXA2-ANXA2R axis known to play a role in cancerogenesis by promoting cell invasion and migration[32,33]. Annexin 2 is the most abundant protein in breast cancer-derived exosomes and enhances angiogenesis[34].

The analysis of the putative enhancer sequence revealed a presence of several transcription factor binding motifs, including motifs for the FOXM1 transcription factor (Supplementary Fig. 6). *FOXM1* expression was significantly increased in malignant gliomas: DA and GB (Fig. 6D). Comparison of *FOXM1* and *ANXA2R* RNA-seq profiles showed a highly significant positive correlation between their expression levels in 33 samples from our cohort (Fig. 6E) and in 299 TCGA glioma samples (Fig. 6F). This high level of co-expression between *FOXM1* and *ANXA2R* indicates a presence of a regulatory network involving these two genes. To further explore the association between their expression levels and the patient's predicted outcome, we performed survival analyses stratifying patients according to *ANXA2R* or *FOXM1* expression levels

(Fig. 6G) using TCGA datasets. For both genes, patients with higher expression levels had shorter survival, even when patients with DA and GB were not combined (Supplementary Fig. 6). This shows that the identified *ANXA2R* regulation may be clinically relevant.

To validate experimentally the discovered regulatory link, we evaluated *ANXA2R* and *FOXM1* expression levels in different glioma cells: LN18, LN229, U87 established glioma cell lines, patient-derived WG4, IPIN glioma cell cultures[35] and normal human astrocytes (NHA) using quantitative PCR (qPCR). Relative to normal human astrocytes, the expression of *ANXA2R* in those cells was elevated only in patient-derived WG4 glioma cells; while *FOXM1* expression was significantly elevated in 4 out of 5 cell lines (Fig. 6H). For further studies we have selected WG4 glioma cells due to the highest *ANXA2R* and *FOXM1* expression in those cells.

In order to test whether the predicted enhancer–promoter contact occurs in the context of glioma cells, we performed a Capture-C assay on WG4 cells, retrieving the gene promoter cleavage fragment with all interacting DNA fragments. The profile obtained by plotting the data as a function of chromosome 5 genomic location (Fig. 6I) shows a pattern similar to the one that can be obtained with 4C-seq. The coverage plot shows enrichment in reads around the studied enhancer denoted by a purple arrow (Fig. 6I), confirming occurrence of the enhancer–promoter contact in glioma cells. As illustrated in Fig. 6J, the results show a significant enrichment with the FOXM1 antibody at the enhancer and promoter of *ANXA2R* (in comparison to the neutral antibody). Moreover, FOXM1 binds to known FOXM1 targets in cancer cells: the *CCNB1* and *PLK1* gene promoters (*CCNB1* encodes CYCLINB1 and *PLK1* encodes Polo kinase 1, respectively) in WG4 cells as demonstrated by ChIP-qPCR. In contrast, there is no specific enrichment with a FOXM1 antibody at the promoters of *H19*, *myoglobin* and *CCND1* 3 which are genes non-expressed or imprinted in glioma cells (they serve as negative controls). These results obtained with Capture-C and validated by ChIP-qPCR confirm the predicted chromosomal contact of FOXM1 in the *ANXA2R* locus. To validate the importance of FOXM1-ANXA2R interactions in glioma pathogenesis, we silenced FOXM1 expression with a FOXM1 specific siRNA (a mixture of four targeting siRNAs). The efficacy of silencing was very good at the mRNA and protein levels (Fig. 6K, L). The expression of *ANXA2R* was significantly reduced in FOXM1-depleted cells when compared to controls. Diffusive growth due to high migratory behavior and invasiveness of tumor cells are hallmarks

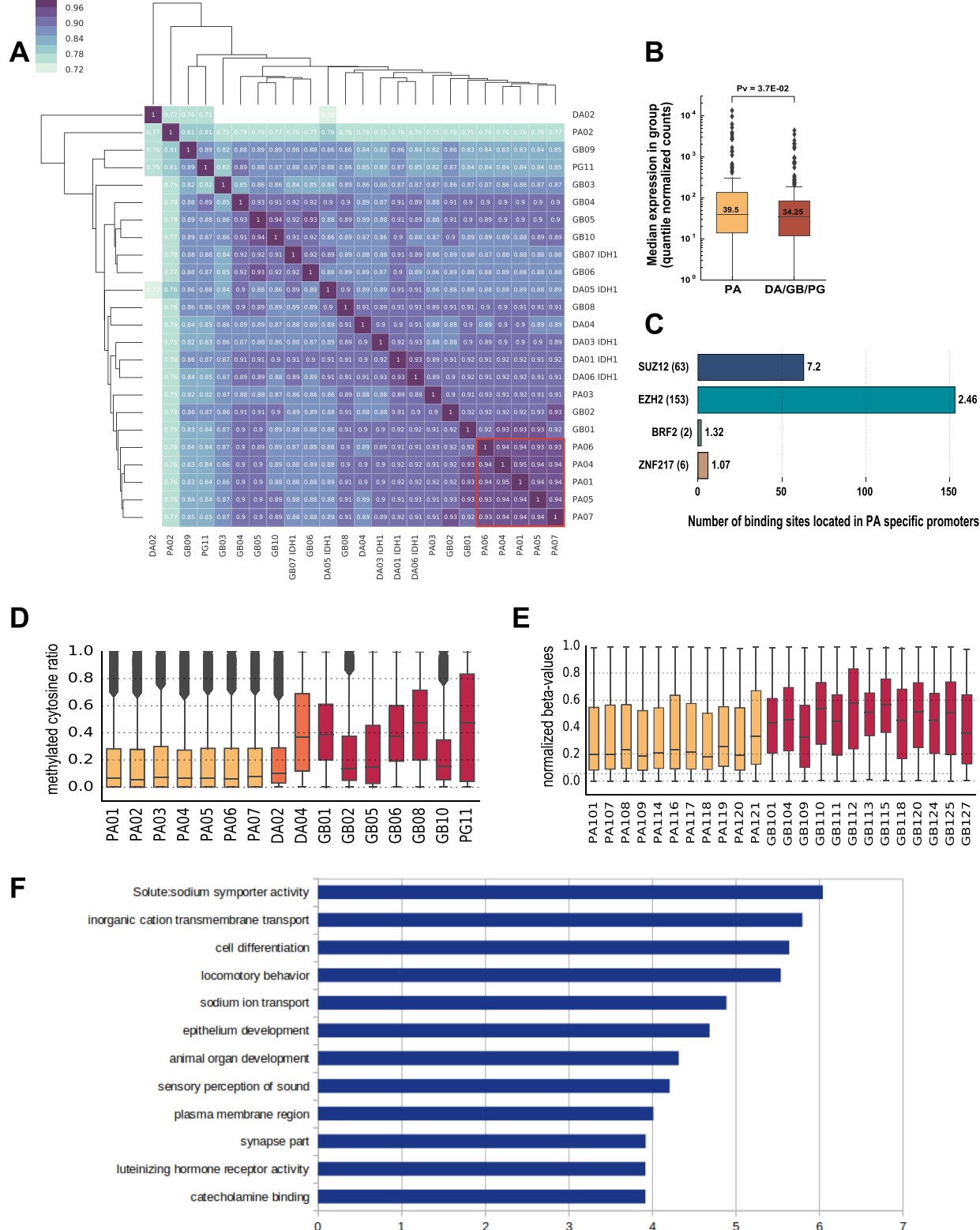

of glioblastoma. We used two assays to evaluate these features of WG4 glioma cells. We demonstrate that both migration (as tested in a scratch assay) and invasion of glioma cells (tested in a Matrigel assay) were significantly reduced in FOXM1-depleted cells with downregulated *ANXA2R* expression (Fig. 6M, N). Altogether, these results support the hypothesis of functional interactions between FOXM1 and the *ANXA2R* regulatory regions in glioma pathogenesis.

## Discussion

Accessible chromatin across the genome reflects a network of permitted interactions through which enhancers, promoters, insulators, and chromatin-binding factors cooperatively regulate gene expression. We combined several methods using massively parallel DNA sequencing to assay chromatin accessibility, active or repressive modification of histones to map regulatory sites instructive for gene expression in gliomas. Gliomas arise from

**Fig. 5 A group of promoters targeted by PRC2 exhibits H3K4me3 hypermethylation and DNA hypomethylation in PAs. A** Hierarchical clustering of samples based on the presence of H3K4me3 at the promoters. Color scale and numbers on the heatmap indicate a similarity between H3K4me3 patterns at the promoters (dark blue, 1 - identical; white, 0 - maximally dissimilar). Red box – the identified cluster of PA samples. **B** Expression of transcripts associated with promoters exhibiting H3K4me3 hypermethylation in PAs. The *p* value was calculated with one-sided Mann–Whitney *U* test. **C** Enrichment of binding sites from the ENCODE transcription factor ChIP-seq data in PAs specific promoters compared to promoters active in all samples. The number of peaks for each DNA-binding protein is given in parentheses; enrichment values per megabase are shown next to each bar and indicated with proportional coloring (red – highest, blue-lowest). **D** DNA methylation in promoters exhibiting H3K4me3 hypermethylation in PAs and having an EZH2 binding site. Methylation levels were determined with WGBS. Samples with *IDH* mutations were excluded from the plot. Colors indicate sample grades (light orange – PA, dark orange – DA, red – GB/PG). **E** DNA methylation in promoters exhibiting H3K4me3 hypermethylation in PAs and having an EZH2 binding site in an independent cohort, determined by HumanMethylation450 BeadChip array. Colors indicate sample grades (light orange – PA, red – GB). **F** GO terms enriched in the set of the PA-specific genes with EZH2 binding sites in promoters. The enrichment was calculated using the PANTHER Overrepresentation Test . More detailed results are provided in Supplementary Fig. 5E. In panels (**B**), (**D**), and (**E**) data are represented as boxplots in which the box shows the quartiles of the dataset, the middle line is the median and the whiskers extends to the largest or smallest value no further than 1.5× the inter-quartile range.

---

neural stem cells, progenitors or dedifferentiated the nervous system cells[36]. It is expected to find active chromatin regions characteristic of the tissue from which tumor originated as well as tumor specific activity in the data obtained from the bulk tumor material. By combining acquired data, we create the Atlas of regulatory sites in gliomas encompassing brain-specific regulatory sites. Within the presented Atlas that describes glioma epigenome features we discriminated two categories of identified regulatory regions: common and variable. The dataset integrating cis-acting regulatory elements which are commonly present in our cohort tend to exhibit the brain-specific signature supported by the evolutionary conservation analysis and intersection with publicly available data for non-brain cells. Taking this into account, our Atlas can serve as a tool not only for assessment of gliomagenesis related events, it can also be beneficial in the field of neuropsychiatric disorders research giving the evidence of a biologically relevant role of sites overlapping with GWAS-identified SNPs[37].

While the utility of the mapped common promoters and enhancers is broader than just its application to the glioma samples, our study has uncovered also a large universe of distal regions that are surprisingly variable in their activity between samples from different glioma grades. As we have shown in this study, many of these, previously unannotated regions can be linked to their likely target genes using Hi-C data from publicly available datasets, obtained from both glioma and fetal brain samples. Such mapping can be very useful in studying biological mechanisms operating in gliomas of different grades, as exemplified in our analysis of the *ANXA2R* enhancer. The putative enhancer sequence showed a presence of several transcription factor binding motifs, including motifs for a FOXM1 transcription factor. The expression of *FOXM1* and *ANXA2R* RNA-seq profiles showed a strong correlation between our samples and in 299 independent TCGA glioma samples. Importantly, their higher expression levels are associated with the shorter patient survival. We validated the interactions using ChIP-qPCR and Capture-C assay. ANXA2-ANXA2R axis is known to play a role in cancerogenesis by promoting cell invasion and migration[32,33]. We demonstrate that both migration and invasion of glioma cells were significantly reduced in FOXM1-depleted cells with down-regulated *ANXA2R* expression. These results show the functional importance of interactions between FOXM1 protein and the *ANXA2R* regulatory regions in gliomagenesis. An inverse correlation of its expression with patient survival indicates clinical relevance.

We interrogated the genome-wide maps of open chromatin and modified histones correlating with transcriptional activity to find regulatory networks distinct in tumors of various grades. We showed significant differences between the H3K27ac and

H3K4me3 profiles in PA and DA/GB samples. We found significantly higher H3K27ac levels in DA/GBs versus PA globally, but a detailed analysis uncovered a group of genes with H3K27ac levels higher in the PAs, contrary to the global trend. Closer inspection of these genes showed that they are characterized by neuronal functions (synaptic transmission and signaling, nervous system development, AMPA receptor activity), reflecting higher contribution of non-transformed neural compartments in PAs. A set of transcription factor binding site motifs indicates potential regulation of these genes by transcription factors such as NRF1, REST, E2F1/4, CTCF, MAX. *NRF1* encodes Nuclear Respiratory Factor 1, a transcription factor that activates the expression of mitochondrial oxidative metabolism genes[38] that are critical for the maintenance of neuronal homeostasis. E2F1 and E2F4 are transcription factors that control numerous target genes, playing a role in cell-cycle progression and apoptosis, with E2F4 acting as a transcriptional repressor[39]. The Myc/Max/Mad network depending on composition of Myc-Max or Mad-Max complexes directs gene-specific transcriptional activation or repression[40]. Unlike higher-grade gliomas, which usually exhibit multiple driver mutations[41], most PAs exhibit a single driver somatic genetic alteration, leading to activation of the MAPK pathway related to FGFR or NTRK2[42] or with rearrangements generating the *KIAA1549-BRAF* fusion oncogene accounting for ~70% of PAs[43]. A recent single cell sequencing of PAs indicates a higher proportion of mature glia-like cells to progenitor-like cells in PAs as compared to other gliomas[44].

In majority of PA samples we identified a set of H3K4me3 marked promoters which are characterized by a significant increase in the transcript levels, H3K27ac and ATAC-seq signal intensity in comparison to DA/GB samples (Fig. 5B, Supplementary Fig. 5). This promoter set is strongly enriched in binding sites for the PRC2 complex components: EZH2 and SUZ12 proteins according to ChIP-seq data from the ENCODE project[18]. These are likely to be differentiation related genes, consistently with the known dysregulation of EZH2 and its regulatory networks in high grade gliomas that are characterized by abundance of dedifferentiated glioma stem cells[45].

Altogether, we present here the first resource integrating so many different types of data on epigenetic and regulatory profiles in gliomas of various grades. We showed several findings based on this resource, and we are confident that this resource will enable even more mechanistic insights leading eventually to future strategies in the treatment of gliomas.

## Methods

All experiments and sequencing were carried out in the same laboratory, therefore, we could avoid possible sequencing artifacts. All datasets were subjected to rigorous quality control according to Encyclopedia of DNA elements (ENCODE) best

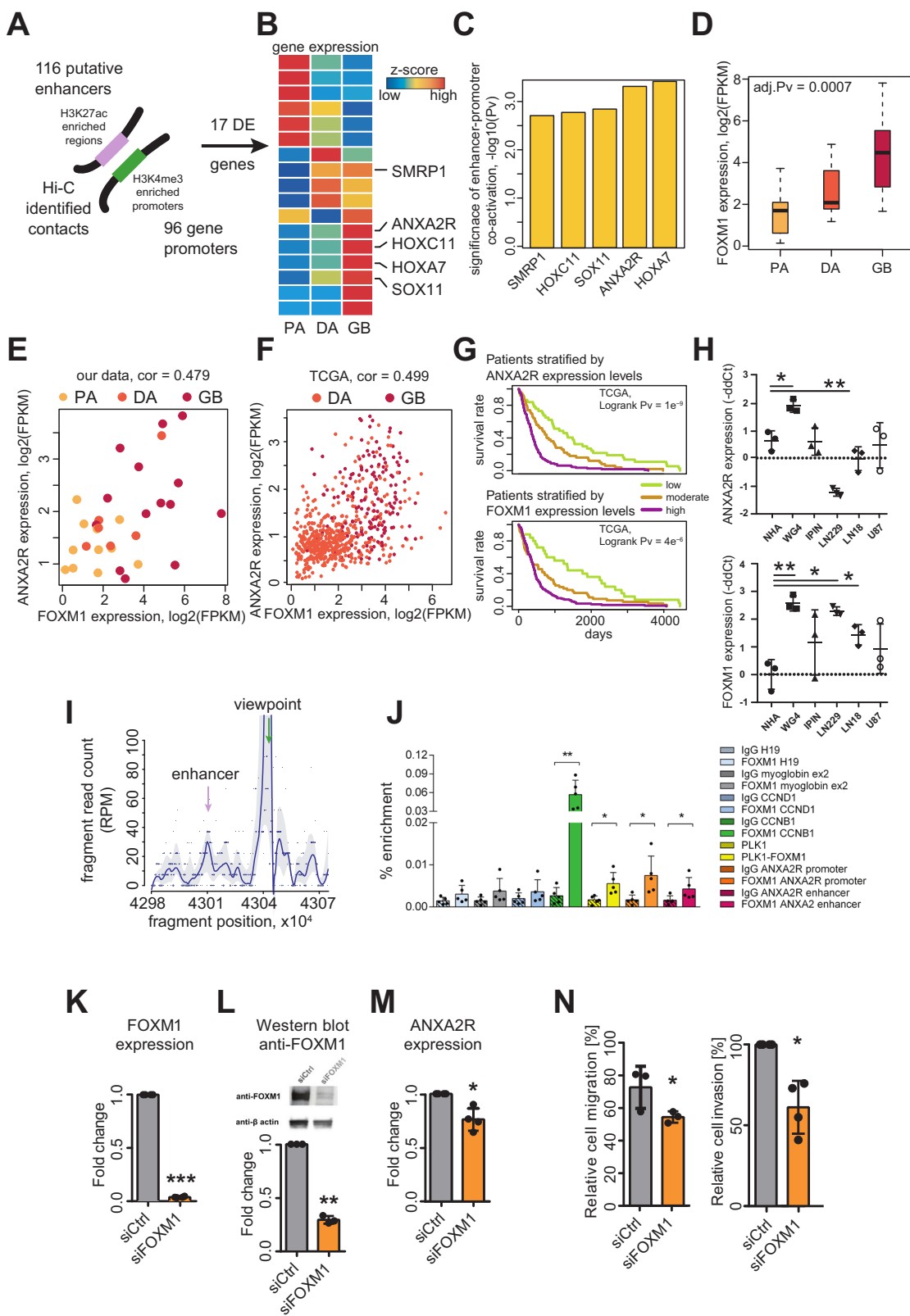

practices, and only datasets passing stringent quality control metrics are presented. For each protocol, the average coverages (normalized read depth in millions [M]) were as follows: for ATAC-seq—76.9 M ($n = 8$), K3K4me3—12.3 M ($n = 24$), H3K27ac—18.7 M ($n = 16$), H3K27me3—41.7 M ($n = 12$), RNAseq—50 M, 25 M per end pairs ($n = 24$), DNA methylation seq 106 M ($n = 24$). Key resources utilized during the experiments and data analysis are listed in Supplementary Table 5.

**Human glioma samples**. Freshly resected glioma specimens were acquired from several neurosurgery clinics: The Children's Memorial Health Institute, Public Central Clinical Hospital, Institute of Psychiatry and Neurology and Mazovian Brodno Hospital. The tissue collection protocol was approved by the Committees of Bioethics of the institutions listed above (protocol number #14/KBE/2012, #KBE/54/2016, #3/2016). Each patient provided a written consent for use of tumor

**Fig. 6 Intersection of the Atlas data with fetal brain Hi-C data reveals FOXM1-ANXA2R axis that impacts survival of GB patients and regulates glioma invasion. A** Graphical representation of chromatin contacts from Hi-C data. **B** Heatmap showing normalized average expression of 17 differentially expressed genes in (FDR < 0.01) selected in the contact analysis. **C** Barplot showing significance of enhancer-promoter co-activation for five genes computed as the Pearson correlation of H3K27ac enrichments in putative enhancers and H3K4me3 enrichments in corresponding promoters. **D** *FOXM1* expression in gliomas. The adjusted *p* values computed with edgeR Bioconductor correspond to expression differences. Logo represents FOXM1 DNA-binding motif. Boxplots show the quartiles of the dataset, lines indicate median and the whiskers the largest or smallest value. **E, F** Scatterplots showing co-expression of *FOXM1* and *ANXA2R* as log-transformed FPKMs in our (**E**) and TCGA (**F**) datasets. Pearson correlation coefficients shown above the plots. **G** Kaplan–Meier survival curves plotted for patients stratified by *ANXA2R* or *FOXM1* expression; differences assessed with the log-rank test. **H** *ANXA2R* and *FOXM*1 expression in normal human astrocytes (NHA), patient-derived (WG4 and IPIN) and established glioma cells (LN229, LN18, U87) determined by RT-qPCR; data normalized to *GAPDH* mRNA; *p* values calculated on the raw data with two-sided *t* test. **I** Capture-C profile representing interactions around the viewpoint in the *ANXA2R* promoter (green arrow). Region enrichments indicate the putative enhancer (purple arrow). Fragment-based raw data are visualized as grey dots, whereas blue dots represent smoothed data. **J** Significant enrichment of FOXM1 binding at the *ANXA2R* promoter and enhancer, and *CCNB1* and *PKL1* promoters (positive controls). No significant FOXM1 enrichment detected at *H19*, *myoglobin* and *CCND1* promoters (negative controls) or with a neutral IgG antibody. Results calculated as % of input, mean ± *SD* (*n* = 5), significance with ratio paired *t* test, two-sided. **K, L** Knockdown of FOXM1 in WG4 glioma cells verified by qPCR and Western blotting; densitometry of immunoblots determined from three experiments, mean ± *SD*, (**K**) two-sided paired *t* test, (**L**) two-sided *t* test. **M** Reduced *ANXA2R* expression in siFOXM1-transfected cells determined with qPCR; mean ± *SD*, n = 4, two-sided paired *t* test. **N** Knockdown of FOXM1 reduces migration and invasion of glioma. Cell migration after scratch quantified in four fields before and 24 h after scratch. Results shown as mean ± *SD*, n = 4. Invasion determined with Matrigel and Transwell® inserts. Images from five fields acquired using fluorescence microscopy, cell nuclei counted with ImageJ software. Results shown as mean ± *SD*, n = 4 (in duplicate). *p* values calculated using paired *t* test, two-sided.

tissues; afterwards samples were anonymized. Details concerning age, gender, clinical diagnosis, tumor location, and molecular test results are provided in the Supplementary Table 1. Tumor samples were transported in DMEM/F-12 medium on ice and processed directly after surgical resection. Procedures were halted at a first safe stop within 1–3 h after acquisition. Tumor specimens were transferred to cold PBS, minced with sterile *scissors* or scalpel on a Petri dish kept on ice and subsequently homogenized by using chilled manual glass douncer. Then the homogenized material was aliquoted to perform different methods.

**Human glioma cell lines**. Human malignant U-87 MG, LN18, and LN229 glioblastoma cells were purchased from American Type Culture Collection (ATCC). WG4 and IPIN are primary glioma cell lines developed from GBM patient surgical samples as described before[35].

Established glioma cell lines were cultured in Dulbecco's modified Eagle medium (DMEM) supplemented with 10% fetal bovine serum (ThermoFisher Scientific) and 100 units/mL of penicillin and 100 μg/mL of streptomycin. Primary glioma cell lines were cultured in DMEM/Nutrient Mixture F-12, GlutaMAX™ medium (DMEM/F-12, GlutaMAX™) supplemented with 10% fetal bovine serum (ThermoFisher Scientific) and antibiotics (100 U/mL penicillin, 100 μg/mL streptomycin).

**Nucleic acids extraction from glioma samples**. Total RNA and DNA were isolated using Tri-Reagent extraction (Sigma–Aldrich, Munich, Germany) starting from 50–100 mg of tissues (depending on the initial specimen size). RNA quality and yield were verified by Bioanalyzer 2100 (Agilent Technologies, Santa Clara, CA) using a RNA 6000 Nano Kit (Agilent Technologies, Ltd.) and NanoDrop 2000 (Thermo Scientific, NanoDrop products, Wilmington, USA). DNA purity was estimated using NanoDrop 2000 (Thermo Scientific, NanoDrop products, Wilmington, USA).

**Analysis of *IDH1/2* mutations**. To check *IDH1/2* status, 20 ng of genomic DNA was amplified by polymerase chain reaction (PCR) with specific primers amplifying the exon 6, in which common *IDH* mutations occur. PCR reaction consisted standard buffer, forward primer 5′-GGATGCTGCAGAAGCTATAA-3′, reverse primer 5′-CATGCAAAATCACATTATTGCC-3′ and DNA polymerase (EURx) in a total volume of 25 μL, and followed denaturation at 95 °C for 30 s, annealing at 54 °C for 30 s and extension at 72 °C for 50 s for 35 cycles. PCR products were separated on 1% agarose gel as 230 bp fragments, and visualized by SimplySafe™ (EURx) staining. Subsequently, PCR products were purified using Syngen GEL/PCR Mini Kit (Syngen Biotech) and a mutational status of *IDH* gene in surgical glioma specimens was determined by Sanger sequencing.

**RNA sequencing**. Strand-specific polyA enriched RNA libraries were prepared using the KAPA Stranded mRNA Sample Preparation Kit according to the manufacturer's protocol (Kapa Biosystems, MA, USA). Briefly, mRNA molecules were enriched from 500 ng of total RNA using poly-T oligo-attached magnetic beads (Kapa Biosystems, MA, USA). Obtained mRNA was fragmented and the first-strand cDNA was synthesized using a reverse transcriptase. Second cDNA synthesis was performed to generate double-stranded cDNA (dsDNA). Adenosines were added to the 3′ ends of dsDNA and adapters were ligated (adapters from NEB, Ipswich, MA, USA). Following the adapter ligation, uracil in a loop structure

of adapter was digested by USER enzyme from NEB (Ipswich, MA, USA). Adapters containing DNA fragments were amplified by PCR using NEB starters (Ipswich MA, USA). Library evaluation was done with Agilent 2100 Bioanalyzer using the Agilent DNA High Sensitivity chip (Agilent Technologies, Ltd.) Mean library size was 300 bp. Libraries were quantified using a Quantus fluorometer and Quanti-Fluor double stranded DNA System (Promega). Libraries were run in the rapid run flow cell and were paired-end sequenced (2x76 bp) on HiSeq 1500 (Illumina, San Diego, CA 92122 USA).

**ATAC-sequencing**. In order to obtain single cell suspension, preliminarily disrupted tumor sample aliquots corresponding to 50–100 mg of tissue were passed through a syringe needle around 50 times. Mechanical homogenization was followed by centrifugation for 5 min at 4 °C at 2,400 g. Each pellet was resuspended in 10 ml of cold lysis buffer L1 (50 mM HEPES KOH, pH7.5, 140 mM NaCl, 1 mM EDTA pH 8.0, 10% glycerol, 5% NP-40, 0.25% Triton X-100, containing proteinase inhibitor cocktail) and incubated at 4 °C for 20 min on a rocking shaker. Then mechanical forcing of tissue disruption was repeated, residual debris were precleared by filtration through an 80-μm nylon mesh filter and eventually lysis buffer was replaced with PBS. Cell suspension was visually controlled under the microscope. Cells were counted automatically with NucleoCounter NC-100 and 50,000 cells were subsequently lysed as previously described[46]. Then transposition reaction was performed using Nextera DNA Library Preparation kit by Illumina, accordingly to the Buenrostro protocol. Reactions were cleaned up with Zymo Clean and Concentrator 5 columns. The remainder of the ATAC-seq library preparation was performed as described previously[46]. Finally, ATAC-seq libraries were visualized on Bioanalyzer 2100 (Agilent Technologies, Santa Clara, CA) and generated chromatograms were used to estimate DNA concentration. Libraries were run in the rapid run flow cell and were paired-end sequenced (2x76 bp) on HiSeq 1500 (Illumina, San Diego, CA 92122 USA).

**Chromatin immunoprecipitation (ChIP) on tissue samples**. Cell suspensions corresponding to 100–400 mg of tumor tissue were aliquoted and spuned down at 1200 rpm (290 × g) for 10 min at 4 °C in a swing bucket centrifuge (Eppendorf Centrifuge 5810 R). Pellets were crosslinked in a crosslinking buffer containing 100 mM NaCl, 50 mM HEPES, 1 mM EDTA, 0.5 mM EGTA and supplemented with 1% formaldehyde (Sigma–Aldrich) for 15 minutes at room temperature on a rocking shaker. Fixation was stopped by incubation with 0.125 M glycine for 5 min at RT. In order to wash away the excess of formaldehyde, fixed material was twice centrifuged at 1,400 g for 10 min at 4 °C and then pellets were resuspended in cold PBS supplemented with a protease inhibitor cocktail (Roche). Eventually, pellets were stored at −80 °C before starting further procedure. Thawed on ice, pellets were resuspended in 5 ml of cold PBS supplemented with proteinase inhibitor cocktail and further homogenized with insulin syringes to obtain single cell suspension (if needed fixed tissue was additionally processed with mechanical homogenizer [PRO Scientific, PRO200] before using syringes). Homogenization step was followed by centrifugation for 5 min at 4 °C at 2,400 g. Each pellet was resuspended in 10 ml of cold lysis buffer L1 (50 mM HEPES KOH, pH7.5, 140 mM NaCl, 1 mM EDTA pH 8.0, 10% glycerol, 5% NP-40, 0.25% Triton X-100, containing a proteinase inhibitor cocktail) and incubated at 4 °C for 20 min on a rocking shaker. Subsequently, ice-cold manual glass douncer was used to release the nuclei and the remaining material was collected by centrifugation for 10 min at 4 °C at 1700 g. L1 buffer was changed for 10 ml of warmed lysis buffer L2 (200 mM

NaCl, 1 mM EDTA pH 8.0, 0.5 mM EGTA pH 8.0, 10 mM Tris pH 8.0, containing proteinase inhibitor cocktail) and incubated at room temperature for 20 min on a rocking shaker, then manual glass homogenizer was used once again. This step was followed by centrifugation for 5 min at 4 °C at 1,700 g. Collected material was resuspended in 0.5 ml of L3 buffer (1 mM EDTA pH 8.0, 0.5 mM EGTA pH 8.0, 10 mM Tris pH 8.0, 100 mM NaCl, 0.1% Na-deoxycholate, 0.17 mM N-lauroyl sarcosine, containing protease inhibitors) and sonicated using a Bioruptor Plus Sonicator (Diagenode) for 3 × 15 cycles (30 ON: 30 OFF) set on HIGH conditions. Lengths of chromatin fragments (200–500 bp) were evaluated on agarose gels. Prior to gel electrophoresis, batches of 10 µl of chromatin were collected by centrifugation at 14,000 rpm (19,283 × g) (Eppendorf Centrifuge 5430) and after adding 1×TE buffer up to 300 µl, samples were incubated at 65 °C with occasional shaking. After reversal of crosslinking, samples were treated with RNase I at 0.1 mg/ml for 30 min at 37 °C, and then treated with proteinase K (final concentration 0.4 mg/ml) for 1.5 h at 55 °C. The DNA was purified by phenol–chloroform extraction followed by ethanol precipitation, and recovered in 20 µl of water. In addition, the efficiency of chromatin isolation was evaluated with the measurement on Quantus Fluorometer (Promega).

DNA-protein complexes were immunoprecipitated with 5 µg of antibody against H3K4me3 (cat. Number 07-473, Merck Millipore), H3K27ac (cat. Number 39133, Active Motif) and H3K27me3 (cat. number ab192985, Abcam). In order to estimate specificity and purity of ChIP reaction, immunoprecipitation with normal IgG was performed simultaneously (cat. Number 2729 S, Cell Signaling). Generally, 30 µg of fragmented chromatin was added per ChIP reaction. Depending on antibody type, different ChIP protocols were applied. Anti-H3K4me3 immunoprecipitation and input sample processing were carried out accordingly to the protocol provided with ChIP-IT Express Immunoprecipitation Kit (Cat. Number 53008, Active Motif) with some modifications: (i) the volume of IP reaction was equal 0.5 or 1 ml, depending on chromatin concentration in L3 buffer, (ii) each washing step of magnetic beads was repeated twice, (iii) elution lasted 30 min. Decrosslink and DNA purification was performed as for sonication test. Anti-H3K27ac and anti-H3K27me3 immunoprecipitation was performed accordingly to the RoadmapEpigenomics protocol for ChIP-seq on tissues (http://www.roadmapepigenomics.org/protocols/type/experimental/). Briefly, 30 µg of pre-cleared chromatin was incubated for 4 h at 4 °C with A- and G-Sepharose beads (Cat. number 16-156 & 16-266, Millipore) mix with pre-bound appropriate antibodies. The beads were washed once with RIPA-150 buffer, twice with RIPA-500, twice with RIPA-LiCl and twice with TE buffer, and then bound chromatin was eluted with freshly made elution buffer. After decrosslink and DNA purification, acquired DNA was measured with Quantus fluorometer using Agilent 2100 Bioanalyzer and High Sensitivity DNA kit (Agilent Technologies, Ltd). Before DNA library preparation, an immunoprecipitated material was evaluated by qPCR with specific primers designed to amplify the GAPDH promoter region (active chromatin marks) or the HOXA7 gene body region (repressive chromatin mark) using SYBR Green chemistry (Cat. Number 4385612, Applied Biosystem by Thermo Fisher Scientific) and QuantStudio 12 K Flex Real-Time PCR System.

**ChIP sequencing**. DNA libraries for chromatin immunoprecipitation with respective antibodies were prepared using QIAseq Ultra Low Input Library Kit (QIAGEN, Hilden, Germany). Briefly, DNA was end-repaired, adenosines were added to the 3′ ends of dsDNA and adapters were ligated (adapters from NEB, Ipswich, MA, USA). Following the adapter ligation, uracil was digested by USER enzyme from NEB (Ipswich, MA, USA) in a loop structure of adapter. Adapters containing DNA fragments were amplified by PCR using NEB starters (Ipswich MA, USA). Library quality evaluation was done with Agilent 2100 Bioanalyzer using the Agilent DNA High Sensitivity chip (Agilent Technologies, Ltd.) Quantification and quality evaluation of obtained samples were done using Nanodrop spectrophotometer (Thermo Scientific, NanoDrop products, Wilmington, USA), Quantus fluorometer (Promega Corporation, Madison, USA) and 2100 Bioanalyzer (Agilent Technologies, Santa Clara, USA). Mean library size was 300 bp. Libraries were run in the rapid run flow cell and were single-end sequenced (65 bp) on HiSeq 1500 (Illumina, San Diego, CA 92122 USA).

**ChIP-qPCR on cultured glioma cells**. Primary glioma WG4 cell cultures were harvested at 90% confluency using standard trypsinization protocol, collected by centrifugation (300 g, 10 min), and fixed in 1% formaldehyde (ThermoFisher Scientific) in PBS for 15 min at room temperature on a rocking shaker. Formaldehyde was quenched by adding 0.125 M glycine and incubating samples for 5 min at RT. Fixed cells were centrifuged at 1,000 g for 10 min at 4 °C and washed with PBS 3 times. Cell pellets were resuspended in 10 ml of NEB buffer (10 mM Tris-HCl pH7.5, 150 mM NaCl, 1 mM EDTA, 1% Igepal, protease inhibitors) and incubated at 4 °C for 60 min on a rotary shaker. The nuclei were collected by centrifugation (10 min, 4 °C, 1,000 g), reconstituted in SDS lysis buffer (1% SDS, 10 mM EDTA, 50 mM Tris-HCl, pH = 8.0) and incubated on ice for 30 min. 1×TE was added to dilute SDS and sonication was performed with Covaris M220 (21′, DF = 25%, PIP = 75, 7.5 W). Sonication efficiency was examined with agarose gel electrophoresis. Chromatin isolated from $5 \times 10^6$ cells was preincubated for 2 hrs with either 5 µg FOXM1 (cat. no. C15410232-100, Diagenode) or normal rabbit IgG antibody (cat. no. PP64B, Merck Millipore). The complexes were then incubated with Dynabeads Protein A (cat. no. 10002D, ThermoFisher Scientific), washed with RIPA-150,

RIPA-500, RIPA-LiCl and TE buffers and eluted with elution buffer (0.1 M NaHCO₃, 0.2% SDS, 5 mM DTT) for 10 min in 65 °C. The elution step was repeated and the eluate was pooled. De-crosslinking, RNase and proteinase K treatments were performed as described in the ChIP on tissue samples. DNA was purified using ZymoReasearch DNA Clean and Concentrator kit (D4003T), according to producent's protocol. Real-time PCR amplifications of the studied ANXA2R regulatory regions were performed applying SYBR Green chemistry (Cat. Number 4385612, Applied Biosystem by Thermo Fisher Scientific) and Quant-Studio 12 K Flex Real-Time PCR System using primers indicated in Key Sources Table. The CCNB1 and PLK1 gene promoters served as positive controls and H19 imprinting control region, myoglobin and CCND1 3 (primers from Ideal ChIP-seq Kit for Transcription Factors, C01010055, Diagenode) were used as negative controls for FOXM1 binding. Results were calculated as % recovery = $2^{\wedge}(Ct\_input-log(input\ dilution;2)-Ct\_IP)*100\%$, mean ± SD ($n = 4$). Paired $t$ test was used for evaluation of significance.

**DNA methylation sequencing**. DNA samples were bisulfite-converted using EZ DNA Methylation-Lightning Kit (Zymo Research, Irvine, CA, USA). Probes from SeqCap Epi CpGiant Enrichment Kit (Hoffmann-La Roche, Basel, Switzerland) were used to enrich each Bisulfite-Converted Sample Library in the predetermined various genome regions of >80.4 Mb capture size comprising >5.6 million of CpG sites on both DNA strands. The libraries were prepared according to the Hoffmann-La Roche's "NimbleGen SeqCap Epi Library Workshop Protocol, v1.0" and "SeqCap Epi Enrichment System User's Guide, v1.2". Briefly, the concentration of genomic DNA was measured using a Quantus fluorometer with QuantiFluor dsDNA System (Promega, Madison, WI, USA) and 1 µg of input DNA together with 165 pg of Bisulfite-Conversion Control (viral unmethylated gDNA; SeqCap Epi Accessory Kit; Hoffmann-La Roche) were fragmented using Focused-ultrasonicator Covaris M220 (Covaris, Inc., Woburn, MA, USA) to an average size of 200 ± 20 bp. DNA fragments were checked using the High Sensitivity DNA Kit on a 2100 Bioanalyzer (Agilent Technologies, Inc., Santa Clara, CA, USA). Next, DNA fragments were "End-Repaired", "A-Tailing" was performed and Index Adapters ligated using KAPA LTP Library Preparation Kit (KAPA Biosystems, Wilmington, USA), SeqCap Adapter Kit A and B (Hoffmann-La Roche) and DNA purification beads (Agencourt AMPure XP Beads; SeqCap EZ Pure Capture Bead Kit; Hoffmann-La Roche). Then, DNA fragments, enlarged by adapters, were size selected with Agencourt AMPure XP Beads (SeqCap EZ Pure Capture Bead Kit) using Solid Phase Reversible Immobilization technology to discard DNA fragments larger than ~450 and smaller than ~250 bp. Next, the libraries were bisulfite-converted using EZ DNA Methylation-Lightning Kit (Zymo Research) and amplified by Pre-Capture Ligation Mediated PCR (LM-PCR). After cleaning on Agencourt AMPure XP Beads (SeqCap EZ Pure Capture Bead Kit), quality and concentrations of Amplified Bisulfite-Converted Sample Libraries were determined using NanoDrop spectrophotometer (Thermo Fisher Scientific, Waltham, MA, USA) and Quantus with QuantiFluor dsDNA System (Promega), respectively. Also a size of DNA fragments was analyzed using the High Sensitivity DNA Kit on a 2100 Bioanalyzer (Agilent Technologies, Inc.). Next, 1 µg of each Amplified Bisulfite-Converted Sample Library was hybridized (47 °C, 67 ± 2 h) with probes from SeqCap Epi CpGiant Enrichment Kit (Hoffmann-La Roche), bound to the Capture Beads (SeqCap EZ Pure Capture Bead Kit; Hoffmann-La Roche) and nonspecifically washed out of contamination and unspecific DNA in buffers of Seq-Cap Hybridization and Wash Kit (Hoffmann-La Roche). Finally, the Captured Bisulfite-Converted Sample Libraries were amplified in Post-Capture LM-PCR, cleaned up using Agencourt AMPure XP Beads (SeqCap EZ Pure Capture Bead Kit) and the Amplified Captured Bisulfite-Converted Sample Libraries were submitted to the last quality check where the quality and the concentrations of the final libraries were determined using NanoDrop (Thermo Fisher Scientific) and Quantus with QuantiFluor dsDNA System (Promega), respectively. A size of the obtained DNA fragments was also analyzed using the High Sensitivity DNA Kit on a 2100 Bioanalyzer (Agilent Technologies, Inc.). Libraries were run in the rapid run flow cell and were paired-end sequenced (2x76 bp) on HiSeq 1500 (Illumina, San Diego, CA 92122 USA).

**DNA methylation bead chip analysis**. Genomic DNA was extracted using the QIAsymphony DNA Midi kit (Qiagen, Crawley, UK). Bisulphite conversion of 500 ng of each sample was performed using the EZ-96 DNA Methylation-Gold kit (Zymo Research, Orange, CA, USA). Bisulphite-converted DNA was used for hybridization on the Infinium HumanMethylation 450 BeadChip[47] (according to manufacturer's instructions). Illumina GenomeStudio software was used to extract the raw signal intensities of each probe (without background correction or normalization).

**Quantitative PCR (qRT-PCR)**. Total RNA was extracted from glioma cells using the RNeasy Mini kit (Qiagen, Hilden, Germany) and purified using RNeasy columns accordingly to the manufacturer's instructions. cDNA was synthetized by extension of oligo(dT) primers with SuperScript III Reverse Transcriptase (Invitrogen, USA). Real-time PCR was performed applying SYBR Green chemistry (Cat. Number 4385612, Applied Biosystem by Thermo Fisher Scientific) on QuantStudio 12 K Flex Real-Time PCR System device using primers indicated in Key Sources

Table. Amplified product was normalized to the endogenous expression of *GAPDH* and represented as minus delta delta Ct values. *P* values were considered significant when *$P < 0.05$ and **$P < 0.01$ (*t* test).

**Capture-C assay**. WG4 cells were cultured, harvested and crosslinked following the same protocol as described in the ChIP-qPCR section. Cell pellet was resuspended in 500 μL of freshly prepared ice-cold lysis buffer (10 mM Tris-HCl, pH 8, 10 mM NaCl, 0.2% Igepal NP-40 (Sigma), 1x protease inhibitor cocktail (Roche)), incubated on ice for 30 min and followed by centrifugation to pellet nuclei (5 min, 1200 rpm/142 × g, 4 °C, Eppendorf Centrifuge 5430). Cells were washed once with 300 μl 1x NEB2 buffer, and nuclei were extracted by 40 min incubation at 37 °C in 190 μl 0.5% SDS 1xNEB2 buffer. In order to quench SDS, samples were transferred on ice and 400 μL of 1X NEBuffer2 and 120 μL of 10% Triton X-100 were added (in the mentioned order), and kept at 37 °C for 15 min. After centrifugation and washing with 1X NEBuffer2, the digestion reaction with 400 U of MboI was performed overnight in 37 °C (300 μl final volume) in termomixer with shake 800 rpm (21 × g). The next day, nuclei were pelleted again, reconstituted in 200 μl 1xNEB2 buffer and additional 200 U of MboI were added for 2 more hours before heat inactivation (65 °C, 15 min). After centrifugation, the obtained pellets were resuspended in 1.2 ml of ligation mix consisting of 1x T4 DNA Ligase buffer, 1% Triton X-100 and 0.1 mg/ml BSA. Eventually 5 μL of 2,000 U/μL T4 DNA Ligase (NEB) was added and incubated at 16 °C for 5 h to perform slow ligation. Then samples were centrifuged and nuclei were resuspended in 200 μl 1X NEBuffer2. Decrosslink was performed by overnight incubation at 65 °C. The next day, we performed RNAse A (10 μL of 10 mg/mL, 30 min, 37 °C) and Proteinase K treatment (20 μL of 20 mg/mL,1.5 h, 55 °C) followed by standard fenol:chloroform extraction to purify DNA. At the same time non-digested and non-ligated controls were processed, as 10 μl of reaction at the crucial steps were taken to check digestion and ligation efficiency by running DNA on 0.8% agarose gel. Properly digested and re-ligated samples were subsequently used for library preparation. First, DNA was fragmented using Diagenode Bioraptor Pico device (cat. No. B01060010). Aliquots of 100 μl DNA (10 μg/μl) were distributed to 6 Eppendorf 1.5 ml tubes and the following conditions: 20"ON/60"OFF, 10 cycles were applied. Sonication efficiency was tested by loading 50 ng DNA on the 1.2% agarose gel. After AMPure beads (Beckman Coulter) purification 1 μg of fragmented DNAs were used for sequencing library generation using NEBNext Ultra Library Prep Kit for Illumina (cat. No. E7370S/L) following a manufacturer's protocol. The library was hybridized to probes capturing the *ANXA2R* promoter fragment. Probe design excluded repeating elements present in the *ANXA2R* promoter region. In order to capture both strands of a selected restriction fragment, biotinylated olignucleotides complementary to 5′ end of each strand were used (IDT). Lyophilized DNA oligonucleotides were reconstituted to a concentration of 2.9 μM, mixed and diluted in equimolar quantities to final 2.9 nM concentration. Subsequently, Hybridization Capture of DNA libraries using xGen Lockdown Probes and Reagents kit (IDT) was used following a manufacturer's instruction with a exception of using NEBNext Ultra Library Prep Kit for Illumina (cat. No. E7370S/L) instead of recommended libarary preparation kit. Final library was sequenced using MiSeq MICRO 2 × 151 bps 300 cyles PE Cycle Paired-end sequencing (4 M reads).

Quality of fastq files was assessed using FastQC tool (https://www.bioinformatics.babraham.ac.uk/projects/fastqc/) and trimmed with FASTQ Trimmer from 3′ end to reach a read length of 81 base pairs. Reads were aligned to human hg38 genome using qAlign function from Bioconductor QuasR package with maxHits parameter set to 1 to obtain uniquely mapped reads. At this point fastq files were treated as files obtained out of single-end sequencing experiments. Generated BAM files were sorted with Picard, and subsequently the same tool (MarkDuplicates option) was used to remove PCR duplicates. We calculated the number of reads mapped to MboI restriction fragments qAlign function (QuasR package). Briefly, new *qProject* object was created out of sorted and deduplicated BAM files, a list of restriction fragments for chromosome 5 generated with getRestrictionFragmentsPerChromosome function (HiTC package) were transformed into query object and mask parameter was set to exclude captured restriction fragment applying vmatchPattern function from Biostrings package. At this point read counts for each file were merged and normalized to "per million reads". Results were visualized with visualizeViewpoint function from Basic4Cseq package.

## Data analysis
*Fastq preprocessing*. The quality of raw fastq files was assessed using FastQC software (https://www.bioinformatics.babraham.ac.uk/projects/fastqc/), and based on the results the following parameters of preprocessing were chosen: (i) for RNA-seq reads were trimmed with FASTQ Trimmer[48] (offset from 5′ end: 11, offset from 3′ end: 3), overrepresented sequences were removed using clip tool from FASTX package (http://hannonlab.cshl.edu/fastx_toolkit/) and reads which pair mate was filtered out were removed; (ii) for ChIP-seq reads were trimmed with FASTQ Trimmer (offset from 5′ end: 5 for samples GB07, GB10, 3 for the rest; offset from 3′ end: 3), Filter FASTQ[48] was used to keep only reads with quality above specified threshold (min quality 10, max number of bases below this quality: 6), for patients DA05 and GB03, in addition, we run Trimmomatic[49] with option "sliding window 4", finally the reads were filtered by length using Filter FASTQ (min. length: 20); (iii) for ATAC-seq reads were trimmed with FASTQ Trimmer

(offset from 5′ end: 14; offset from 3′ end: 3), Filter FASTQ was used to keep only reads with quality above specified threshold (min quality 10, max number of bases below this quality: 5), the reads were filtered by length using Filter FASTQ (min. length: 20) and reads which pair mate was filtered out were removed. All fastq preprocessing steps were performed using Galaxy wrappers[50].

*Mapping, peak calling, read counts*. ChIP-seq and ATAC-seq reads were mapped to hg38 using Bowtie 2.2.6.2[51] with default parameters. RNA-seq reads were mapped to hg38 transcriptome using Tophat2[52], with default parameters, except for the following: library type = fr-firststrand, mate-std-dev = 100, mate--inner-dist = 300. For ChIP-seq and ATAC-seq duplicates were removed using samtools rmdup command[53]. ChIP-seq peaks were called using MACS2.1[54], with default parameters. For ATAC-seq, F-seq1.85[55] with default parameters was used, except for f = 50. RNA-seq counts on genes and transcripts were calculated using samtools. Mapping and peak-calling with MACS were performed using Galaxy wrappers.

*Defining active promoters and putative enhancers*. Promoters were defined as region +/−1,000 bp around every TSS. We presumed that a promoter has a H3K4me3 mark in patient A if it overlaps with any H3K4me3 peak called in a patient A; a gene has a H3K4me3 mark if any of its promoters has a H3K4me3 mark. Similar presumptions were made for H3K27ac marks and ATAC-seq signals. Overlaps were found using bedtools intersect tool[56]. For every patient non-promoter H3K27ac peaks were defined as peaks that have no overlap with regions +/−1,000 bp from any TSS. Putative enhancers were defined by merging all non-promoter peaks identified for each patient (individual peak files were merged into a single file and overlapping peaks were merged into one peak using bedtools merge tool). Each putative enhancer was denoted as active in a patient A if it overlapped with any non-promoter H3K27ac peak present in a patient A.

*Common and variable marks*. Active genes, promoters and enhancers were divided into two groups: common, which are present in at least threshold patients, and variable, which are present in less than threshold patients. Threshold value was chosen separately for every mark/type of genomic element. For promoters and genes with H3K4me3 mark, threshold = 20. For promoters and genes with H3K27ac mark, threshold = 10. For enhancers with H3K27ac mark, threshold = 4.

*Coverage*. Coverages over whole genome were calculated using bedtools genomecov tool. Coverages over specific intervals were calculated using bedtools intersect tool and custom python script calculate_coverage_over_features.py. Briefly, for every interval sum of coverages over all positions were calculated; then the sum was divided by length of the interval to obtain mean coverage value. Normalized coverages were obtained by dividing every coverage value by mean coverage defined as summaric coverage (the sum of values on every position of the genome, including ones with no coverage) divided by the length of the genome.

*Evolutionary conservation analysis*. A collection of random genomic intervals was generated for both common active promoters and common active enhancers using bedtools shuffle. PhastCons 100-Way scores were obtained from http://hgdownload.cse.ucsc.edu/goldenPath/hg38/phastCons100way/hg38.phastCons100way.bw Mean PhastCons scores for all analyzed intervals were calculated with bigWigAverageOverBed from the kentUtils package (https://github.com/ENCODE-DCC/kentUtils). One-sided Mann–Whitney *U* test was calculated to assess significance of difference in conservation scores between identified regulatory elements and random genomic intervals.

*Identification of enhancer-promoter contacts based on Hi-C data*. Hi-C data for fetal brain samples from[13] were used to infer long-range chromatin interactions for putative enhancers defined based on the presence of non-promoter H3K27ac peaks. First, coordinates of putative enhancers were converted from hg38 to hg19 using UCSC liftover tool, to match the Hi-C data. Then the HiCEnterprise method (https://github.com/regulomics/HiCEnterprise) was used to identify contacts within 2 Mb distance with 10 kb resolution[57]. Coordinates of identified contacts were converted back to hg38.

*Correlation between histone marks on promoters and transcript expression*. H3K4me3 and H3K27ac ChIP-seq coverages on promoter regions and RNA-seq counts on transcripts were normalized with quantile normalization. Only transcripts of protein-coding genes were considered. For genes with multiple alternative transcripts, the transcript with a highest ChIP-seq coverage on promoter was chosen. Spearman correlation between ChIP-seq coverage and transcript expression was calculated for all transcript-promoter pairs in all patients. For violin plot visualization, transcripts were divided into quantiles based on expression level (quantile normalized counts).

*Correlation between H3K27ac level on enhancers and transcript expression*. For each enhancer identified by the presence of H3K27ac mark, the closest transcript was found using bedtools closest. The enhancers were subsequently divided into three groups: (i) having the closest transcript within less than 20 kilobases, (ii) having

closest transcript within 20–200 kb, (iii) having closest transcript located >200 kb away. Spearman correlation between H3K27ac coverage and transcript expression was calculated for each of these groups of enhancers. Then, enhancers assigned to each group were checked for contacts with promoters, predicted from Hi-C data, and Spearman correlation was again calculated for enhancer-transcript pairs defined with Hi-C. In cases where one enhancer had predicted contacts with more than one promoter, mean of counts for transcripts associated with all contacting promoters was used in the correlation calculation. $P$ value for the difference between correlation obtained for closest enhancer-transcript pairs and pairs defined with Hi-C data was estimated by random sampling 260 enhancer-transcript pairs from the set of 3,371 enhancer-transcript pairs with closest transcript located further than 200 kb away for 1,000 times.

From all enhancer–promoter pairs predicted based on Hi-C data pairs with significant correlation between H3K27ac coverage and transcript expression were chosen based on the following criteria: FDR < 0.1, Spearman R > 0.7, minimal read counts on transcript in a sample with highest expression = 20. Amongst analyzed pairs, 116 enhancer–promoter pairs fulfilled these criteria.

*Clustering based on gene expression.* RNA-seq row counts on genes of 37 cases (4 NB, 11 PA, 7 DA, 14 GB, 1 PG) were plotted for each tumor grade and hospital of a sample origin. No batch effect related to grade or hospital was detected[58]. Two normalization procedures were considered following recommendations:[58] (i) within-lane to adjust for GC-content and gene-length; (ii) between-lane, both implemented in EDASeq 3.8 R package[58]. Genes with mean row counts across all samples below 10 were filtered out, and then the EDASeq 3.8 protocol was applied with full-quantile normalization method for within- and between-lane normalization. The filtering step resulted in 19,328 genes that were used for clustering. Dendextend R package[59] was used to perform hierarchical clustering with ward.D2 method with distance set to Euclidean. ssGSEA tool (gene-centric single sample Gene Set Enrichment Analysis of gene expression data) was used to define GB subtypes (downloaded from https://github.com/broadinstitute/ssGSEA2.0) Input data for the analysis consisted 37 samples and 56,819 RPKM normalized gene expression values, converted to GCT file format. In addition, four reference files (for each subtype separately) have been downloaded from Molecular Signatures Database v7.0 (http://software.broadinstitute.org/gsea/msigdb/index.jsp) and used in the analysis. To predict the subtype of Pan Glioma RNA Expression Cluster (LGr1, LGr2, LGr3, LGr4), a validation cohort from published studies, including 665 adult and pediatric gliomas[1] and 21,022 of their gene expressions levels were compiled. Next on that data we trained three classifiers (Naïve Bayes, SVM and Random Forest) and applied them as a committee of classifiers with voting mechanism on our 26 samples to predict the target variable (Pan Glioma Cluster).

Gene expression differential analyses were computed with edgeR Bioconductor package. For visualization and co-activation analyses FPKM (Fragments per Kilobase Million) values were calculated.

*Comparison of ChIP-seq and ATAC-seq profiles in TSS-proximal regions.* Genomic positions with the numbers of mapped reads above the significance threshold of $z$ score = 7 were identified as anomalous, and the tags mapped to such positions were discarded. Read frequencies were computed in 500-bp non-overlapping bins for each sample independently and normalized by the corresponding library sizes to represent values per one million of mapped reads. For the histone marks abundance was computed per bin as a difference between normalized frequencies obtained from chip and input experiments. For each TSS proximal region, the abundance values were computed as an average over abundance in bins overlapping the corresponding 4 kb region (TSS +/−2 kb). The transcription start site proximal regions were defined as +/−2 kb from the TSS. TSS proximal regions overlapping with other genes were excluded from this analysis. The averaged and detailed profiles around TSS were computed using 10 bp non-overlapping bins and smoothed using runmean function from caTools package with k = 15.

*Clustering based on H3K4me3 mark.* For each sample a set of promoters with H3K4me3 mark was defined based on a presence of H3K4me3 ChIP-seq peak as described above. Pairwise similarities between samples were defined as Jaccard indices calculated for sets of promoters with H3K4me3 mark. These pairwise similarities were subsequently used to perform hierarchical clustering of samples with scipy.cluster.hierarchy.linkage function and the UPGMA algorithm (method = "average").

*Identification of promoters specifically active/inactive in the PA cluster.* To identify promoters whose activity differentiate PA from higher grade glioma, for each annotated promoter we compared the number of samples from the PA cluster in which it was active (marked with H3K4me3) to the number of samples from a group of 13 higher grade samples (DA01, DA03, DA04, DA05, DA06, GB03, GB04, GB05, GB06, GB07, GB08, GB09, GB10) in which it was active. As specifically active in PA, we chose those active in at least four out of five samples from the PA cluster and in at most two higher grade samples and additionally having $p$ value from Fisher's exact test below 0.003. Similarly, promoters specifically active in higher grades were defined as those active in at most one PA sample and at least 11 higher grades samples, and having the $p$ value below 0.003.

*Enrichment analysis for DNA-binding proteins binding sites.* The ENCODE Regulation "Txn Factor" track data was downloaded from http://hgdownload.soe.ucsc.edu/goldenPath/hg19/encodeDCC/wgEncodeRegTfbsClustered/. Promoters specifically active in PAs, active in all samples or inactive in any sample were intersected with bed files defining binding sites of 161 proteins using bedtools intersect. Enrichments in binding sites were calculated as a fold change of the number of intersections between each protein binding sites and promoters specifically active in PAs and number of intersections for the same protein binding sites and active or inactive promoters (per megabase).

*Protein binding motif analysis.* To identify transcription factors that can potentially bind to the identified promoters with H3K27ac hyperacetylation or H3K4me3 hypermethylation in PA, we used HOCOMOCO-v11 motif database[60]. For hyperacetylated promoters the relative motif enrichment was computed with AME tool from MEME-Suite software (v5.0.4)[61] and logos were obtained from the HOCOMOCO website. For hypermethylated promoters, motif enrichment was computed with PWMEnrich R package[62] using sequences of promoters inactive in all samples as background. The returned protein binding motifs were accepted if their $p$ value was below 0.001. To establish the similarity of motifs enriched for grade-specific groups (PAs vs. higher grade gliomas - HGGs) on each of the backgrounds, the Jaccard index (Jaccard, 1901) implemented in "jaccard" CRAN R package[63] was used.

*Analysis of enriched Gene Ontology terms.* Enrichment of gene ontology terms for promoters with H3K4me3 hypermethylation in PAs and EZH2 binding sites was performed using the PANTHER Overrepresentation Test (Released 20171205)[64,65] and Fisher's Exact with FDR multiple test correction. For promoters with H3K27ac hyperacetylation in PAs, gene ontology analysis was performed with GO.db Bioconductor package and 0.05 significance threshold on Bonferroni corrected $P$ value (Fisher's exact test) was used.

*DNA methylation data processing.* For DNA methylation, assessment of methylation sites was restricted to regions covered by SeqCap Epi CpGiant Methylation panel which captures 80.5 Mb (Roche). The methylation analysis workflow consisted of: read quality assessment, mapping to hg38 genome, removal of PCR duplicates, coverage statistics assessment and methylation level assignment. The software required to perform the aforementioned steps: FastQC, BSMAP[66], Picard (http://broadinstitute.github.io/picard/) were selected based on Roche recommended pipeline[67] and compiled into a single CytoMeth tool (https://github.com/mdraminski/CytoMeth) enabling fast and transparent rough data processing. The following quality requirements were set: minimal bisulphite conversion 99.2% and at least 3,000,000 cytosines in a CpG context covered with at least ten reads. Out of 26 samples, 20 passed these requirements and were used to assess DNA methylation variability within promoter regions (+/−2,000 bp from TSS) with a use of methylKit[68].

For Infinium HumanMethylation 450 BeadChip data signal intensities were imported into R using the methylumi package[69]. Initial quality control checks were performed using functions in the methylumi package. The methylation score for each CpG was represented as a $\beta$ value according to the fluorescent intensity ratio representing any value between 0 (unmethylated) and 1 (completely methylated). Mapping of CpGs was performed based on UCSC Genome Browser annotation data[70]. CpGs beta values in the range from 0 to 0.2 were defined as hypomethylated and those from 0.8 to 1 as hypermethylated.

*DNA methylation analysis for PAs specific promoters with EZH2 binding sites.* Signal intensities for probes from HumanMethylation450 BeadChip array, which were located within PA specific promoters with EZH2 binding sites, were compared for 11 independent PAs and 13 GB samples. DNA methylation data from previously pubslihed studies[26] and gene expression data from the ltater were downloaded from Gene Expression Omnibus (GEO: http://www.ncbi.nlm.nih.gov/geo/) using the following accession numbers: GSE61431, GSE44684, and GSE44971, respectively. Quantile normalization was performed on signal intensities. For the expression data median intensities were calculated for each probe in each analyzed group of samples.

*TCGA data analyses.* The expression values for the TCGA samples were downloaded from TCGA website (RNASeqV2 set). Survival analyses were performed with TCGA data. The patients were stratified into three equinumerous sub-groups according to *ANXA2R* or *FOXM1* expression levels. Correlation between the expression level of the selected genes and patient survival times was analyzed using Kaplan–Meier model and log-rank test.

**Reporting summary**. Further information on research design is available in the Nature Research Reporting Summary linked to this article.

## Data availability

Data that support the findings of this study is available at our resource website: http://regulomics.mimuw.edu.pl/GliomaAtlas and have been deposited in the European

Nucleotide Archive with the accession code **ERP125425**. A file with the source data underlying the Figures and Supplementary Figures is available at http://regulomics.mimuw.edu.pl/GliomaAtlas/manuscript_supplement/Source_data_files.zip Source data are provided with this paper.

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

## Acknowledgements

This work was supported by Polish National Science Center grant [DEC-2015/16/W/NZ2/00314] (to BW), [UMO-2018/28/T/NZ2/00510] (to KS) and by the Foundation for Polish Science TEAM-TECH Core Facility project "NGS platform for comprehensive diagnostics and personalized therapy in neuro-oncology" (BW, BG). We would like to thank Marcin Kruczyk for his contributions to the concept of the study, Luca Giorgetti and Mariya Kryzhanovska for their help in performing the Capture-C experiments and Jacek Koronacki for comments on the paper. Fig. 1 has been partially created with BioRender.com.

## Author contributions

Conceptualization: J.K., B.K., M.J.D., K.D., B.Wi., K.St. Methodology: B.K., K.St. Investigation in the lab: K.St., R.G., S.K., B.G., K.P. Computational investigation: B.Wo., M.M., A.Mac., J.M., A.D., I.G., H.K., K.Si., M.D., M.J.D., M.J., B.Wi. Resources: K.K., W.G., M.R., T.C., A.Mar. Writing—Original draft: K.St., J.M., M.M., B.K., B.Wi. Writing—Review & editing: B.Wi., B.K., K.S., M.J.D., M.M., K.D., J.K. All authors have read and approved the final version of the paper.

## Competing interests

The authors declare no competing interests.
