## [Peer Review File · Nature Communications]

REVIEWER COMMENTS

Reviewer #1 (Remarks to the Author):

The manuscript by Stepniak et al provides an important characterization of epigenomic profiles in low- and high-grade gliomas. The authors generated ChIP-seq datasets for key histone post-translational modifications, mapped chromatin accessibility using ATAC-seq, and performed RNA-seq and DNA methylation analyses using primary surgical specimens. They integrated this information with previously published Hi-C datasets generated from brain tissue to infer regulatory networks in gliomas. Data integration led to the identification of an active FOXM1-ANXA2R axis that might contribute to the biology of gliomas.

The role of FOXM1 in gliomagenesis is not new and extensive literature exists on this topic (see for instance PMIDs 28344040, 22014570, 25601206, 23991102, etc), but its proposed regulation of ANXA2R appears new, to the best of my knowledge. In any case, the convergence of epigenomic information on FOXM1 is confirmation that the datasets generated by Stepniak et al are biologically relevant and could inform new discoveries.

I have some reservations regarding 2 of the 4 main claims of this submission:

(1) "We provide an atlas of cis-regulatory elements active in human gliomas." I am not sure the number of samples investigated for each glioma grade is sufficient to define the overall datasets as an atlas.

(2) "Chromatin loop activates FOXM1-ANXA2R pathological network in glioblastomas."

Pathogenicity of this axis was not experimentally validated in in vivo models and therefore this conclusion is not warranted based on the existing data.

MAJOR ISSUES

- I was not able to find information on how tumor-infiltrating cells (particularly the ones of hematopoietic origin) were removed from the surgical samples prior to performing ChIP and ATAC. This is crucial information because it will provide a measure of the confounding effects of non-tumor cell types on the epigenomic datasets generated.

- After tumor dissociation, was cell viability scored (by any method) before using the cells for ChIP?

- How deeply were the libraries sequenced?

- I was not clear on how many samples were successfully profiled with full epigenomic datasets (ie how many were profiled for K27ac, K4me3, K27me3, RNA-seq and ATAC-seq)? This is not indicated and would be an important complement to Figs 2A and 2B.

- I was not able to find information on how many replicates were generated for each dataset from the same sample. As you know, there is always some variability in ChIP, ATAC and RNA-seq output, and replicates are an absolute requirement for this type of work, especially when the number of samples used is not large.

- The correlation between samples in Fig 5A is very good (usually >0.80), irrespective of tumor classification. In view of this, I would like to know how much of variation observed between epigenomic datasets is patient-specific and how much is disease-specific. Basically, how much of the variance is explained by inter-personal differences, and how much is actually due to the epigenetic of each disease?

- PRC2 usually has a repressive role on chromatin accessibility and activity and its enrichment at active enhancer sites is paradoxical. Since you have RNA-seq datasets for the same samples, could you check if genes encoding PRC2 members are downregulated in these tumors (PAs vs GBM, for instance)? If this is the case, then downregulation of the PRC2 complex could explain activity at these enhancers. Some ChIP experiments on primary samples and primary cell cultures for PRC2 members should be performed.

- Fig 6I: The enrichment of putative interactions between the bait and the enhancer (marked by the arrow) is not higher than at other regions. The authors should use multiple control (positive and negative) bait regions for this type of experiment to show specific enrichment of the predicted interactions.

- For the ChIP-qPCR described in Fig 6J, the enrichment of FOXM1 at the enhancer element of ANXA2R is modest. It is standard for ChIP-qPCR experiments to use multiple positive and negative

control regions. Negative control regions are particularly important to determine the overall level of background globally.

- On a similar note, the test of the two antibodies against FOXM1 should have at least some error bars and some statistics to be able to assess enrichment. Negative controls again are needed to assess specificity.

MINOR ISSUES

- In the Introduction, the Fontebasso et al reference specifically deals with pediatric gliomas, which are molecularly different from adult gliomas. This reference should be substituted with the Brennan et al (TCGA paper) reference, which found that about half high-grade glioma cases have at least one mutation in epigenetic/chromatin regulators.

- I was not able to find Table S1. I apologize in advance if I just missed it.

- There is a mis-spelling in Fig 2A: GD instead of GB.

- Better definition and context of LGr classification should be provided.

- Figs 5D-E: The y-axis label "Ratio" is not clear. Could you define better?

- Could you add a map of the relevant genomic region in Fig 6I?

Reviewer #2 (Remarks to the Author):

In the manuscript entitled "Mapping chromatin accessibility and active regulatory elements reveal new pathological mechanisms in human gliomas" the authors performed several analyses in glioma samples claiming that represents the most complete molecular and epigenetic characterization on tumor samples ever made.

Starting from different types of glioma, they performed analysis of whole genome profiles of open chromatin, histone modifications, DNA methylation and gene expression. This led to a more precise mapping of active promoters and enhancers in gliomas, some reflecting brain specific sites. In addition, they claimed that approach revealed a new regulatory connection between FOXM1 and ANXA2R implicated in gliomagenesis. They concluded that they provide an atlas of brain specific enhancers and promoters which could be a valuable resource for further exploration.

In general, I do believe the work proposed by Stępnik et al. represents a complete molecular analysis of glioma analysis. They do provide a good resource to the field in particular to whom are interested in understating gene regulation in human brain tumor. Nevertheless, the authors should provide more data about FOXM1/ANXA2R interaction.

Main Comments:

- The intensive analysis of the glioma samples, differing from type and aggressiveness, represents a valuable resource for the entire glioma field. In particular, the parallel analysis of the same tumor from different points of view using several techniques gives a more global and precise regulatory and epigenetic profiles at once.

- My main concern is on the last part where the authors analysed FOXM1 and ANXA2R interaction. Here the impact on survival is only theoretical and based on their co-expression in more malignant brain tumors and a positive correlation with the survival of the patients. Despite their possible interaction that has been shown in cell lines it does not mean there is an impact on malignancy. I would strongly suggest the authors to provide more functional data. For example, despite the effect of FOXM1 knockdown is well demonstrated to have a negative impact on glioma cell lines proliferation, it would be worth to test whether the shRNA for FOXM1 (with control shRNA and also data that knocking down is effective) in glioma cells have an impact on ANXA2R expression. In addition, I would suggest also to analyse for cell invasion and migration since the authors stated in the discussion that ANXA2-ANXA2R axis known to play a role in cancerogenesis by promoting cell invasion and migration (D'Souza et al., 2012; Shiozawa et al., 2008). Using a shANXA2R would be appreciated too.

It would be also worth discussing the possible explanation about the up-regulation of both FOXM1 and ANXA2R being observed only in one cell line. How similar or dissimilar those lines are?

I would suggest rephrasing the sentence "This approach revealed a new regulatory connection between FOXM1 and ANXA2R implicated in gliomagenesis" because the provided data do not support that.

Minor comments and suggestions

- Suggestion to move to the methods section the part "All experiments and sequencing were carried out in the same laboratory, therefore, we could avoid possible sequencing artifacts. All datasets were subjected to rigorous quality control according to Encyclopedia of DNA elements (ENCODE) best practices, and only datasets passing stringent quality control metrics are presented".

- Results paragraphs 1 and 2 can be fused.

REVIEWER COMMENTS

Reviewer #1 (Remarks to the Author):

The manuscript by Stepniak et al provides an important characterization of epigenomic profiles in low- and high-grade gliomas. The authors generated ChIP-seq datasets for key histone post-translational modifications, mapped chromatin accessibility using ATAC-seq, and performed RNA-seq and DNA methylation analyses using primary surgical specimens. They integrated this information with previously published Hi-C datasets generated from brain tissue to infer regulatory networks in gliomas. Data integration led to the identification of an active FOXM1-ANXA2R axis that might contribute to the biology of gliomas.

The role of FOXM1 in gliomagenesis is not new and extensive literature exists on this topic (see for instance PMIDs 28344040, 22014570, 25601206, 23991102, etc), but its proposed regulation of ANXA2R appears new, to the best of my knowledge. In any case, the convergence of epigenomic information on FOXM1 is confirmation that the datasets generated by Stepniak et al are biologically relevant and could inform new discoveries.

Ad. We are grateful for high evaluation of the work presented in our study.

I have some reservations regarding 2 of the 4 main claims of this submission:

(1) “We provide an atlas of cis-regulatory elements active in human gliomas.” I am not sure the number of samples investigated for each glioma grade is sufficient to define the overall datasets as an atlas.

Ad. The presented dataset is the largest (in gliomas) set of integrated open chromatin, histone modifications, DNA methylation and transcriptomic data from the same cancer patient, it is unique not only for gliomas. We agree that a number of samples is not sufficient to speak of an atlas of cis-regulatory elements for each grade, therefore we focus on differences between low grades and malignant gliomas. We have changed the wording in the text to make it clearer.

(2) “Chromatin loop activates FOXM1-ANXAR2 pathological network in glioblastomas.” Pathogenicity of this axis was not experimentally validated in in vivo models and therefore this conclusion is not warranted based on the existing data.

Ad. We have provided new results verifying that the FOXM1-ANXAR2 pathological connection operates in glioma cells (new Fig. 6K-N). Our data show that the effective silencing of FOXM1 downregulates expression of ANXAR2 and reduces glioma cell migration (in a scratch assay) and invasion (in a Matrigel assay). These results experimentally validate the pathogenicity of this axis in human glioma cells, as high motility and invasiveness are

pathological hallmarks of malignant gliomas.

Figure 6

MAJOR ISSUES

- I was not able to find information on how tumor-infiltrating cells (particularly the ones of hematopoietic origin) were removed from the surgical samples prior to performing ChIP and ATAC. This is crucial information because it will provide a measure of the confounding effects of non-tumor cell types on the epigenomic datasets generated.

Ad. Indeed, the reviewer is right about accumulation of immune cells in gliomas and activation of glial cells in a tumor microenvironment, in particular in GBs. There were two main reasons why tumor-infiltrating cells (particularly the ones of hematopoietic origin) were not removed from the surgical samples. Firstly, removal of hematopoietic cells would require myelin removal, Percoll centrifugation and removal of CD45+ cells by passing cell mixture through columns or by FACS sorting. Those procedures would take additional several hours and we were concerned that such manipulations would affect epigenetic marks (particularly histone modifications) and further reduce a number of cells subjected to analyses. Secondly, the transcriptomic profiles used for glioma diagnostics/subtyping are based on bulk tumor sequencing and in fact, discriminating a mesenchymal GBM subtype is heavily based on expression of stromal genes, including immune cell genes. As we sought to find epigenetic profiles corresponding to the discernible transcriptomic profiles in bulk tumors, it seemed logical to analyze epigenetic profiles in bulk tumors.

- After tumor dissociation, was cell viability scored (by any method) before using the cells for ChIP?

Ad. After each tumor dissociation cell suspension was evaluated under a phase contrast microscope to ensure that a procedure generated a single cell suspension. We could see single, bright cells, typical for live cells. During establishment of this step of the procedure we verified a ratio of live/death cells using an automated system and >90% were live cells.

- How deeply were the libraries sequenced? How many samples were successfully profiled with full epigenomic datasets (ie how many were profiled for K27ac, K4me3, K27me3, RNA-seq and ATAC-seq)? This is not indicated and would be an important complement to Figs 2A and 2B.

Ad. In the revised manuscript we have included information about depth of sequencing in a specific dataset. It is revised as follows: “For a specific protocol an average coverage

(normalized read depth in millions [M]) were as follows: for ATAC-seq- 76.9 M (n=8), H3K4me3-12.3 M (n=24), H3K27ac - 18.7 M (n=16), H3K27me3 – 41.7 M (n=12), RNAseq – 50 M, 25 M per end pairs (n=24), DNA methylation seq 106 M (n=24)”.

Unfortunately, not all samples were successfully profiled for all epigenomic features. This is partly due to a limited size of the biological material available for analysis (especially in the case of samples from pediatric brain tumors). Nonetheless, we have 16 samples with RNA-Seq, H3K27Ac and H3K4Me3 profiles available simultaneously. We have added a supplementary table S2 including all this information.

- I was not able to find information on how many replicates were generated for each dataset from the same sample. As you know, there is always some variability in ChIP, ATAC and RNA-seq output, and replicates are an absolute requirement for this type of work, especially when the number of samples used is not large.

Ad. We did not make technical replicates for a specific tumor sample. All protocols have been applied after first optimizing these conditions in LN18 glioma cells with technical replicates. A survey of recent papers (i.e. Buenrostro et al. Nat Methods. 10(12): 1213–1218, 2013, doi: 10.1038/nmeth.2688; Nat Commun. 2018; 9: 4020, 2018, doi: 10.1038/s41467-018-06258-2) shows that in most studies technical replicates of clinical samples were not performed due to scarcity of the material and reagent costs. Only in studies presenting one protocol i.e. ATACseq in different cancers (including LGG and HGG) in the TCGA study technical replicates were performed (*Corces MR et al. Science Advances 2018*).

In our study, a number of cells and quantity of material isolated from low grade tumors (PA) was insufficient to perform all desired histone modifications in the same individual. In cases of DA or GBM we isolated sufficient quantities of the material to carry on all analyses (including DNase-I seq which was not included), but not to perform technical replicates. The strength of this study stems from integration of numerous datasets for the same individual and we focus on data that internally support each other. The dataset presented in this study is currently the largest dataset of integrated data for gliomas of different grades and a number of samples was considerable. We should stress that all wet lab procedures as well as sequencing were performed by the same researchers to minimize variability.

- The correlation between samples in Fig 5A is very good (usually >0.80), irrespective of tumor classification. In view of this, I would like to know how much of variation observed between epigenomic datasets is patient-specific and how much is disease-specific. Basically, how much of the variance is explained by inter-personal differences, and how much is actually due to the epigenetic of each disease?

Ad. Indeed, the H3K4me3 mark location on promoters is very consistent between samples, what is shown not only in figure 5A but also in figure 3A. Gene expression, ATAC-seq signal and H3K27ac marks at the promoters are also relatively consistent between samples, while H3K27ac at the enhancers and H3K27me3 at the promoters vary highly between samples. Results of sample clustering based on both H3K4me3 on promoters (Figure 5A) and based on gene expression (Figure 3E) suggest that variation between epigenomic datasets is highly

related to division of samples into PA and higher grade gliomas, but not necessarily to classification into DA and GB. Clustering based on the presence of H3K27ac at the enhancers (Figure 2) suggests even higher contribution of inter-personal differences than disease/grade specific features into the observed variation.

Figure 1. Hierarchical clustering of samples based on the presence of H3K27ac mark at the common non-promoter regulatory regions (active in at least 4 samples). Color scale and numbers on the heatmap indicate similarity between H3K27ac pattern at the non-promoter regulatory regions (dark blue, 1 - identical; white, 0 - maximally dissimilar).

- PRC2 usually has a repressive role on chromatin accessibility and activity and its enrichment at active enhancer sites is paradoxical. Since you have RNA-seq datasets for the same samples, could you check if genes encoding PRC2 members are downregulated in these tumors (PAs vs GBM, for instance)?

If this is the case, then downregulation of the PRC2 complex could explain activity at these enhancers. Some ChIP experiments on primary samples and primary cell cultures for PRC2 members should be performed

Ad. We have described a group of promoters characterized by the presence of active chromatin mark (H3K4me3) preferentially in PAs. We explored the data from the ENCODE project (results of ChIP-seq experiments for transcription factors, performed on several cell lines) to search for factors which might preferentially bind these promoters. As a result we

found that this group of promoters is enriched in putative binding sites of the PRC2 complex when compared to promoters activated in all our samples. This observation suggests that PRC2 is involved in regulation of expression of the genes exhibiting differential promoter activity between PAs and higher grade gliomas. Indeed, we found higher expression of PRC2 complex members in samples from higher grade gliomas (Figure 2) and this difference in expression could influence the activity of this group of promoters.

Figure 2. Expression of EZH2 and SUZ12 genes (the most expressed transcripts) in PAs and higher grade gliomas.

We agree that ChIP experiments on primary samples and primary cell cultures for PRC2 members could be very informative. Unfortunately, there are no commercially available cell lines from low grade gliomas. We and others have tried to established patient derived cell lines from PA and failed.

- Fig 6I: The enrichment of putative interactions between the bait and the enhancer (marked by the arrow) is not higher than at other regions. The authors should use multiple control (positive and negative) bait regions for this type of experiment to show specific enrichment of the predicted interactions.

Ad. In case of 3-C numerous controls are necessary. In a Capture C method during preparation of libraries each DNA fragment is tested against another fragment, in short each fragment of DNA is a control. Capture –C is all-versus-all technique and the experiment was performer in the laboratory and under a guidance of Dr. Luca Giorgetti, a leading expert in this technique (Mol Cell 2020;77(4):688-708, doi: 10.1016/j.molcel.2019.12.021. In this type of experiment, there is a search for a specific enrichment of the predicted interactions. We do not want to claim that it is the only place interacting with the promoter, we just show with another method that the predicted interaction takes place in the studied cells. This observation together with ChIP-qPCR results support our claim that FOXM1 interacts with the enhancer element of *ANXA2R*.

- For the ChIP-qPCR described in Fig 6J, the enrichment of FOXM1 at the enhancer element

of ANXA2R is modest. It is standard for ChIP-qPCR experiments to use multiple positive and negative control regions. Negative control regions are particularly important to determine the overall level of background globally.

- On a similar note, the test of the two antibodies against FOXM1 should have at least some error bars and some statistics to be able to assess enrichment. Negative controls again are needed to assess specificity.

Ad. We performed the requested ChIP-PCR experiments again using one FOXM1 antibody and including 3 negative controls (genes non-expressed or imprinted in glioma cells). The results are presented in a new figure and a statistical analysis of the data shows a significant enrichment with the FOXM1 antibody for enhancer and promoter of ANXA2R, as well as two known promoters (in comparison to the neutral antibody). We agree that the observed enrichment with the FOXM1 antibody for enhancer and promoter of ANXA2R are modest, nevertheless they are specific and significant. Modest enrichment reflects either biological low binding or imperfect quality of antibody. While studying kinetics of silencing of FOXM1 in glioma cells, we observed that its level was reduced with the time of cell culture under standard conditions (with serum).

MINOR ISSUES

- In the Introduction, the Fontebasso et al reference specifically deals with pediatric gliomas, which are molecularly different from adult gliomas. This reference should be substituted with the Brennan et al (TCGA paper) reference, which found that about half high-grade glioma cases have at least one mutation in epigenetic/chromatin regulators.

Ad.

We have added the Brennan et al reference in the introduction

- I was not able to find Table S1. I apologize in advance if I just missed it.

Ad. It was an erroneous omission on our part. The table has been included in the resubmission.

- There is a mis-spelling in Fig 2A: GD instead of GB.

Ad. The figure has been corrected.

- Better definition and context of LGr classification should be provided.

Ad. We are not sure what the reviewer requests. According to the World Health Organization (WHO) classification, WHO grade I and II gliomas are considered low-grade. These two

subcategories are distinct histologically and are classified by a neuropathologist. In the Table S1 we state that all LGG samples (n=11) used in this study were IDH wild type pilocytic astrocytomas which are benign tumors.

- *Figs 5D-E: The y-axis label "Ratio" is not clear. Could you define better? Ad.*

Ad. We have changed axis labels into more informative: "methylated cytosine ratio" in figure 5D (this is the ratio of methylated cytosines to all cytosines detected in a particular site by whole genome bisulfite sequencing) and "normalized β -values" in figure 5E.

- *Could you add a map of the relevant genomic region in Fig 6I?*

Ad. Figure 6 contains already many panels and is very busy. Therefore, we decided not to present a map of the relevant genomic region for Capture -C. The information is provided in the text.

Reviewer #2 (Remarks to the Author):

In the manuscript entitled "Mapping chromatin accessibility and active regulatory elements reveal new pathological mechanisms in human gliomas" the authors performed several analyses in glioma samples claiming that represents the most complete molecular and epigenetic characterization on tumor samples ever made.

Starting from different types of glioma, they performed analysis of whole genome profiles of open chromatin, histone modifications, DNA methylation and gene expression. This led to a more precise mapping of active promoters and enhancers in gliomas, some reflecting brain specific sites. In addition, they claimed that approach revealed a new regulatory connection between FOXM1 and ANXA2R implicated in gliomagenesis. They concluded that they provide an atlas of brain specific enhancers and promoters which could be a valuable resource for further exploration.

In general, I do believe the work proposed by Stepniak et al. represents a complete molecular analysis of glioma analysis. They do provide a good resource to the field in particular to whom are interested in understating gene regulation in human brain tumor. Nevertheless, the authors should provide more data about FOXM1/ANXA2R interaction.

Main Comments:

- The intensive analysis of the glioma samples, differing from type and aggressiveness, represents a valuable resource for the entire glioma field. In particular, the parallel analysis of the same tumor from different points of view using several techniques gives a more global and precise regulatory and epigenetic profiles at once.

- *My main concern is on the last part where the authors analysed FOXM1 and ANXA2R interaction.*

Here the impact on survival is only theoretical and based on their co-expression in more malignant brain tumors and a positive correlation with the survival of the patients. Despite their possible interaction that has been shown in cell lines it does not mean there is an impact on malignancy. I would strongly suggest the authors to provide more functional data. For example, despite the effect of FOXM1 knockdown is well demonstrated to have a negative

impact on glioma cell lines proliferation, it would be worth to test whether the shRNA for *FOXM1* (with control shRNA and also data that knocking down is effective) in glioma cells have an impact on *ANXA2R* expression. In addition, I would suggest also to analyse for cell invasion and migration since the authors stated in the discussion that *ANXA2-ANXA2R* axis known to play a role in cancerogenesis by promoting cell invasion and migration (D'Souza et al., 2012; Shiozawa et al., 2008). Using a sh*ANXA2R* would be appreciated too.

Ad. We have provided new results verifying that *FOXM1-ANXA2R* pathological interaction operates in human patient-derived glioma cells (Fig. 6K-N). Our data show that silencing of *FOXM1* downregulates expression of *ANXA2R*, and reduces glioma cell migration (in a scratch assay) and invasion (in a Matrigel assay). These results experimentally validate the pathogenicity of this axis in human glioma cells, as high motility and invasiveness are hallmarks of malignant gliomas.

Figure 6

We have tried to test also contribution of *ANXA2R* in glioma cells by determining migration and invasion in si*ANXA2R* transfected cells, but two available *ANXA2R* antibodies tested were not specific and we could not verify the effects of silencing.

It would be also worth discussing the possible explanation about the up-regulation of both FOXM1 and ANXA2R being observed only in one cell line. How similar or dissimilar those lines are?

Ad. All tested cell lines originate from glioblastomas, the difference is WG4 and IPIN are patient-derived cell lines and other are commercial, established cell lines from ATTC. Expression of *FOXM1* is increased in a majority of tested cells, but *FOXM1* and *ANXA2R* co-expression was detected in WG4 cells which are patient-derived cells and proliferate more vigorously than IPIN cells. When we studied the kinetics of silencing of *FOXM1*, we observed that its basal levels was reduced with time of culture (after 48-72 h). This observation suggests that *FOXM1-ANXA2R* axis may operate in bulk tumor and in primary glioma cultures, not exposed to differentiating conditions (serum containing media). Three tested established glioma cell lines differ in molecular background (i.e. TP53, PTEN).

I would suggest rephrasing the sentence “This approach revealed a new regulatory connection between FOXM1 and ANXA2R implicated in gliomagenesis” because the provided data do not support that.

Ad. We believe that new data about the importance of FOXM1-ANXA2R axis in glioma migration and invasion (presented in a new figure 6) validate experimentally the concept and support the statement. Also, we have rewritten substantial parts of the manuscript making the context of this sentence more clear.

Minor comments and suggestions

- Suggestion to move to the methods section the part “All experiments and sequencing were carried out in the same laboratory, therefore, we could avoid possible sequencing artifacts. All datasets were subjected to rigorous quality control according to Encyclopedia of DNA elements (ENCODE) best practices, and only datasets passing stringent quality control metrics are presented”.

Ad. We have moved this part to the “Star Methods” document.

- Results paragraphs 1 and 2 can be fused.

Ad. The paragraphs have been merged.

REVIEWERS' COMMENTS

Reviewer #1 (Remarks to the Author):

I am satisfied with the revisions. The interactions measured in Figures 6I and 6J are not strong, but they appear significant and the authors do not overstate their findings.

Please check for inconsistencies in naming ANXA2R (eg ANX2R) in the last Results section.

Reviewer #2 (Remarks to the Author):

The authors addressed the points and concerns raised by the reviewer.

RESPONSE TO REVIEWERS

We are very happy that the reviewers found our previous responses adequate. We have reviewed all the references to the ANXA2R gene to the correct form as per the Reviewer #1 request.

REVIEWERS' COMMENTS

Reviewer #1 (Remarks to the Author):

I am satisfied with the revisions. The interactions measured in Figures 6I and 6J are not strong, but they appear significant and the authors do not overstate their findings.

Please check for inconsistencies in naming ANXA2R (eg ANX2R) in the last Results section.

Reviewer #2 (Remarks to the Author):

The authors addressed the points and concerns raised by the reviewer.